# UNCERTAINTY QUANTIFICATION VIA STABLE DISTRIBUTION PROPAGATION

**Felix Petersen**[1]**, Aashwin Mishra**[1]**, Hilde Kuehne**[2,3]**, Christian Borgelt**[4]**, Oliver Deussen**[5]**,**
**Mikhail Yurochkin**[3]    [1]Stanford University, [2]University of Bonn, [3]MIT-IBM Watson AI Lab,
[4]University of Salzburg, [5]University of Konstanz,    `mail@felix-petersen.de`

## ABSTRACT

We propose a new approach for propagating stable probability distributions through neural networks. Our method is based on local linearization, which we show to be an optimal approximation in terms of total variation distance for the ReLU non-linearity. This allows propagating Gaussian and Cauchy input uncertainties through neural networks to quantify their output uncertainties. To demonstrate the utility of propagating distributions, we apply the proposed method to predicting calibrated confidence intervals and selective prediction on out-of-distribution data. The results demonstrate a broad applicability of propagating distributions and show the advantages of our method over other approaches such as moment matching.[1]

## 1 INTRODUCTION

Neural networks are part of various applications in our daily lives, including safety-critical domains such as health monitoring, driver assistance, or weather forecasting. As a result, it becomes important to not only improve the overall accuracy but also to quantify uncertainties in neural network decisions. A prominent example is autonomous driving [1], [2], where a neural network is not only supposed to detect and classify various objects like other cars or pedestrians on the road but also to know how certain it is about this decision and to allow, e.g., for human assistance for uncertain cases. Such prediction uncertainties can arise from different sources, e.g., model uncertainties and data uncertainties [3]–[6]. These types of uncertainties have received considerable attention over the years. There is a line of work focusing on out-of-distribution (OOD) detection [7], [8], e.g., via orthogonal certificates [9] or via Bayesian neural networks [10]–[15]. Another line of works focuses on quantifying data uncertainties, e.g., via uncertainty propagation [16]–[20], ensembles [21], quantile regression [9], [22], post-hoc recalibration [23], or also via Bayesian neural networks [24]. Data uncertainties can primarily arise from two important sources: (i) input uncertainties, i.e., uncertainties arising in the observation of the input features, and (ii) output uncertainties, i.e., uncertainties arising in the observation of the ground truth labels. In this work, we primarily focus on quantifying how input uncertainties lead to prediction uncertainties, but also introduce an extension for quantifying input and output uncertainties jointly by propagating through a Probabilistic Neural Network (PNN).

To address this task, the following work focuses on propagating distributions through neural networks. We consider the problem of evaluating $f(x + \epsilon)$, where $f$ is a neural network, $x$ is an input data point, and $\epsilon$ is a random noise variable (stable distribution, e.g., Gaussian). To this end, we approximate $f(x + \epsilon)$ in a parametric distributional form, which is typically a location and a scale, e.g., for the Gaussian distribution, this corresponds to a mean and a covariance matrix. Evaluating the true output distribution $Y$ ($f(x + \epsilon) \sim Y$) exactly is intractable, so the goal is to find a close parametric approximation, ideally $\min_{\mu_y, \Sigma_y} D\big(\mathcal{N}(\mu_y, \Sigma_y), Y\big)$ for some distributional metric or divergence $D$.

Due to the complexity of neural networks, specifically many non-linearities, it is intractable to compute the distribution of $f(x + \epsilon)$ analytically. A straightforward way to approximate location and scale, i.e., the mean and (co-)variances, $\mu$ and $\Sigma$, of $f(x + \epsilon)$ is to use a Monte Carlo (MC) estimate. However, this approach can often become computationally prohibitive as it requires evaluating a neural network $f$ a large number of times. Additionally, the quality of the approximation for the covariances as well as the quality of their gradients deteriorates when the data is high-dimensional, e.g., for images. We, therefore, focus on analytical parametric approximations of $f(x + \epsilon)$.

---

[1]The code is publicly available at github.com/Felix-Petersen/distprop.

This work proposes a technique, Stable Distribution Propagation (SDP), for propagating stable distributions, specifically Gaussian and Cauchy distributions, through ReLU-activated neural networks. Propagating the locations (e.g., means for the case of Gaussian) through a layer corresponds to the regular application of the layer or the neural network. For propagating scales (e.g., (co-)variances for the case of multivariate Gaussians), we can exactly propagate them through affine layers and further propose a local linearization for non-linear layers. The motivation for this is as follows: While it is intractable to find an optimal $\mu_y, \Sigma_y$, the problem can be separated into layer-wise problems using a divide-and-conquer strategy as, e.g., also done in moment matching approaches. While divide-and-conquer works, looking at the problem from the perspective of multiple layers and deep networks again shows that any errors in individual layers caused by respective approximations accumulate exponentially. Therefore, the proposed method also considers a more robust approximation. Namely, we rely on the total variation (TV) to find the distribution that maintains as much probability mass at exactly correct locations as possible to reduce the overall propagation error. As the argmax-probability under respective distributions is intractable, we further propose a pairwise distribution loss function for classification tasks. Our evaluation for multiple-layer architectures confirms that this shows a better performance compared to marginal moment matching for multi-layer nets.

Overall, the perspective of $f(x + \epsilon)$ allows assessing the sensitivity of the neural network for uncertainty quantification. In particular, it allows assessing the effects of uncertainties due to input measurement errors (Figure 1, left). Further, incorporating the sensitivity of a neural network during training can also enable selective prediction: because the method is able to regularize or minimize the output variance, the variance of predictions for observed data is

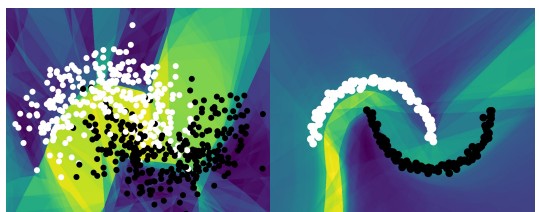

Figure 1: Data (left) and model (right) uncertainty estimation. Yellow / blue indicate high / low uncertainty.

minimized while the variance for predictions of unobserved data tends to be larger (Figure 1, right).

Stable Distribution Propagation (SDP) provides an efficient technique for propagating distributions with covariances (or correlated scales) that performs (i) competitively to moment matching approximations (DVIA) at a fraction of the computational cost and (ii) superior to covariance-free marginal moment matching approaches. To our knowledge, our method is the first method to enable propagation of Cauchy distributions, or more generally the family of symmetric $\alpha$-stable distributions. To empirically validate SDP, we (i) compare it to other distribution propagation approaches in a variety of settings covering total variation (TV) distance and Wasserstein distance; (ii) compare it to other uncertainty quantification methods on 8 UCI [25] regression tasks; and (iii) demonstrate the utility of Cauchy distribution propagation in selective prediction on MNIST [26] and EMNIST [27].

## 2 RELATED WORK

Predictive uncertainties in deep learning can be decomposed into different components: model and data uncertainty. Model uncertainties arise due to model limitations such as variability in neural network weights and biases. Data uncertainties originate from incipient noise in the feature space that is propagated through the model as well as, separately, noise in the target space measurements. The focus of our investigation lies on data uncertainties.

While analyzing the effects of input space uncertainties on predictive uncertainty, prior authors have focused on analytic approaches that avoid sampling. Such investigations address the approximate propagation of normal distributions at individual training samples through the network to model the network response to perturbed inputs [18], [20], [28], [29]. [28] propose Natural-Parameter networks, which allow using exponential-family distributions to model weights and neurons. Similarly, [30], [31] use uncertainty propagation to sampling-free approximate dropout, and [32] extend moment matching for Bayesian neural networks. These methods use (marginal) moment matching for propagating parametric probability distributions through neural networks, as we will subsequently discuss in greater detail. Deterministic Variational Inference Approximation (DVIA) [33] is an approximation to (full) moment matching of multi-variate Gaussian distributions for ReLU non-linearities.

As an alternative, some authors have also focused on the output space uncertainty estimation, using approaches that yield both a point prediction along with the predictive uncertainty. [21] utilize PNNs [34] where the network outputs both the predicted mean and variance, which they augment via adversarial training and ensembling. Contrastingly, [9] learn conditional quantiles for estimating data uncertainties. We use several methods from this category as baselines in our example applications.

Our proposed approach falls into the *first* category and is most similar to [20] and [18], but differs in how distributions are passed through non-linearities such as ReLUs. Previous work relies on (marginal) moment matching [35], also known as assumed density filtering. Moment matching is a technique where the first two moments of a distribution, such as the output distribution of ReLU (i.e., the mean and variance) are computed and used as parameters for a Gaussian distribution to approximate the true distribution. As it is otherwise usually intractable, moment matching techniques match the distribution to a marginal multivariate Gaussian (i.e., with a diagonal covariance matrix) at each layer. Accordingly, we refer to it as marginal moment matching. SDP propagates stable distributions by an approximation that minimizes the total variation (TV) distance for the ReLU activation function. We find that this approximation is faster to compute, better with respect to TV and Wasserstein distances, and allows for efficient non-marginal propagation. Further, as it does not rely on moment matching, it can also be used to propagate Cauchy distributions for which the moments are not finitely defined. Another advantage of SDP is that it can be applied to pre-trained models. In contrast, moment matching changes the mean, causing deviations from regular (point) propagation through the network, leading to poor predictions when applied to pre-trained networks.

Beyond UQ, SDP was inspired by propagating distributions through logic gate networks [36], optimizers [37], simulators [38], and algorithms [39], e.g., for sorting [40], clustering [41], and rendering [42].

## 3   STABLE DISTRIBUTION PROPAGATION

Stable distributions are families of parametric distributions that are closed under addition of random variables (RV) and constants, as well as multiplication by positive constants. To give a concrete example, the sum of two Gaussian distributions is a Gaussian distribution. For Gaussian and Cauchy distributions, the reproductive property also holds under subtraction by RVs and multiplication by negative constants. Therefore, we will focus on Gaussian and Cauchy distributions; however, in Supplementary Material A.2, we provide special consideration of general $\alpha$-stable distributions for positive-weight networks as well as uncertainty bounds for real-weight networks. We emphasize that, throughout this work (with exclusion of SM A.2), we do not put any restrictions on network weights.

For propagating stable distributions, particularly Gaussian and Cauchy distributions, through neural networks, we consider affine transformations and non-linearities, such as ReLU, separately. For the first case, affine transformations, an exact computation of the parametric output distribution is possible due to the reproductive property of Gaussian and Cauchy distributions. For the second case, non-linearities, we propose to use local linearization and (i) show that, for individual ReLUs, this approximation minimizes the TV distance, and (ii) empirically find that, also in more general settings, it improves over substantially more expensive approximations wrt. both Wasserstein and TV.

### 3.1   AFFINE TRANSFORMATIONS

The outputs of fully connected layers as well as convolutional layers are affine transformations of their inputs. We use the notation $\boldsymbol{y} = \boldsymbol{x}\boldsymbol{A}^\top + \boldsymbol{b}$ where $\boldsymbol{A} \in \mathbb{R}^{m \times n}$ is the weight matrix and $\boldsymbol{b} \in \mathbb{R}^{1 \times m}$ is the bias vector with $n, m \in \mathbb{N}_+$. We assume that any covariance matrix $\boldsymbol{\Sigma}$ may be positive semi-definite (PSD) and that scales may be $\geq 0$. These are fairly standard assumptions, and we refer to Dirac's $\delta$-distribution for scales of $0$.

As convolutional layers can be expressed as fully connected layers, we discuss them in greater detail in Supplementary Material A.3 and discuss only fully connected layers (aka. affine / linear layers) here. We remark that AvgPool, BatchNorm, etc. are also affine and thus need no special treatment.

Given a _multivariate Gaussian distribution_ $\boldsymbol{X} \sim \mathcal{N}(\boldsymbol{\mu}, \boldsymbol{\Sigma})$ with location $\boldsymbol{\mu} \in \mathbb{R}^{1 \times n}$ and PSD scale $\boldsymbol{\Sigma} \in \mathbb{R}^{n \times n}$, $\boldsymbol{X}$ can be transformed by a fully connected layer via $\boldsymbol{\mu} \mapsto \boldsymbol{\mu}\boldsymbol{A}^\top + \boldsymbol{b}$ and $\boldsymbol{\Sigma} \mapsto \boldsymbol{A}\boldsymbol{\Sigma}\boldsymbol{A}^\top$.

Given a _multivariate marginal Gaussian distribution_ $\boldsymbol{X} \sim \mathcal{N}(\boldsymbol{\mu}, \text{diag}(\boldsymbol{\sigma}^2))$ with $\boldsymbol{\mu} \in \mathbb{R}^{1 \times n}$, $\boldsymbol{\sigma}_{\geq 0} \in \mathbb{R}^{1 \times n}$, the marginal Gaussian distribution of the transformation of $\boldsymbol{X}$ can be expressed via $\boldsymbol{\mu} \mapsto \boldsymbol{\mu}\boldsymbol{A}^\top + \boldsymbol{b}$ and $\boldsymbol{\sigma}^2 \mapsto \boldsymbol{\sigma}^2 (\boldsymbol{A}^2)^\top$ where $\cdot^2$ denotes the element-wise square.

Given a _multivariate marginal Cauchy distribution_ $\boldsymbol{X} \sim \mathcal{C}(\boldsymbol{x}_0, \text{diag}(\boldsymbol{\gamma}^2))$ with $\boldsymbol{x}_0 \in \mathbb{R}^{1 \times n}$, $\boldsymbol{\gamma} \in \mathbb{R}_{\geq 0}^{1 \times n}$, the marginal Cauchy distribution of the transformation of $\boldsymbol{X}$ is $\boldsymbol{x}_0 \mapsto \boldsymbol{x}_0 \boldsymbol{A}^\top + \boldsymbol{b}$ and $\boldsymbol{\gamma} \mapsto \boldsymbol{\gamma} \, \text{Abs}(\boldsymbol{A})^\top$.

Notably, in all cases, the location $\boldsymbol{\mu}/\boldsymbol{x}_0$ coincides with the values propagated in conventional network layers, which will also generally hold throughout this work. This allows applying this method directly to pre-trained neural networks. In Section 3.3, we discuss how to efficiently and effectively compute the output scales $\boldsymbol{\Sigma} / \boldsymbol{\gamma}$. Now, we cover propagating distributions through non-linearities.

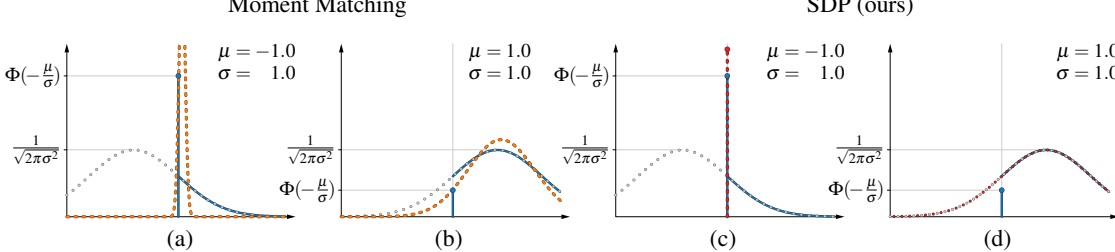

Figure 2: Visualization of two sample parametric approximations of ReLU for moment matching ((a) and (b), orange, dashed) as well as for the proposed Stable Distribution Propagation ((c) and (d), red, dashed). The gray (dotted) distribution shows the input and the blue (solid) is the true output distribution (best viewed in color).

## 3.2 NON-LINEAR TRANSFORMATIONS

To handle non-linearities, we utilize local linearization and transform the locations and scales as:

$$(\mu, \sigma) \mapsto (f_a(\mu), |f_a'(\mu)| \cdot \sigma), \qquad (\boldsymbol{\mu}, \boldsymbol{\Sigma}) \mapsto (f_a(\boldsymbol{\mu}), f_a'(\boldsymbol{\mu}) \, \boldsymbol{\Sigma} \, f_a'(\boldsymbol{\mu})^\top). \qquad (1)$$

As ReLU is the most common non-linearity for neural networks, we give it our primary consideration. Using local linearization, the proposed approximation for transforming distributions with ReLUs is

$$\text{ReLU} : (\mu, \sigma) \mapsto \begin{cases} (\mu, \sigma) & \mu \geq 0 \\ (0, 0) & \text{otherwise} \end{cases} \qquad (2)$$

for distributions parameterized via $\mu$ and $\sigma$. In fact, for stable distributions, this approximation is optimal wrt. TV. Recall that the TV between the true distribution $\boldsymbol{Q}$ and an approximation $\boldsymbol{P}$ is

$$TV(\boldsymbol{P}, \boldsymbol{Q}) = \sup_A \left| \int_A (p - q) \, d\nu \right| = \frac{1}{2} \int |p - q| \, d\nu. \qquad (3)$$

TV is a vertical distance, which is closely related to the KL divergence, i.e., it measures for each point how densities or probability masses deviate. Due to the nature of the ReLU, the KL divergence is undefined (because $\boldsymbol{Q}$ and $\boldsymbol{P}$ have different support.) In the experiments in Section 3.4, we also consider approximation quality wrt. the Wasserstein distance, i.e., a horizontal probability distance. In the following, we formalize that the approximation of a ReLU non-linearity minimizes the TV. Recall that when parameterizing a Gaussian or Cauchy distribution with $(0, 0)$, we are referring to Dirac's $\delta$-distribution. The transformed distributions and their propagations are illustrated in Figure 2.

**Theorem 1.** *Local linearization provides the optimal Gaussian approximation of a univariate Gaussian distribution transformed by a ReLU non-linearity with respect to the total variation:*

$$\underset{(\tilde{\mu}, \tilde{\sigma})}{\arg\min} \, TV(\boldsymbol{P}, \boldsymbol{Q}) = \begin{cases} (\mu, \sigma) & \mu \geq 0 \\ (0, 0) & \textit{otherwise} \end{cases} \qquad (4)$$

$$\textit{where } \boldsymbol{P} = \mathcal{N}(\tilde{\mu}, \tilde{\sigma}^2), \; \boldsymbol{Q} = \text{ReLU}(\mathcal{N}(\mu, \sigma^2))$$

*Proof Sketch.* Recall that by using a stable parametric distribution we can either have a density (i.e., no mass at any point) or a point mass at a single point. Whenever we have a positive location $\mu \geq 0$ (as in Figures 2 (b) and (d)), SDP exactly covers the true probability distribution except for the point mass at 0, which cannot be covered without choosing a point mass. Any deviation from this, like, e.g., the moment matching result leads to an increase in the TV. Whenever we have a negative location $\mu < 0$ (as in Figures 2 (a) and (c)), SDP chooses to use a point mass at 0; any other approximation has a larger TV because more than 0.5 of the mass lies at point 0 in the true distribution.

The proofs are deferred to Supplementary Material A.1.

**Corollary 2.** *Theorem 1 also applies to Cauchy distributions parameterized via $x_0, \gamma$ instead of $\mu, \sigma$.*

After covering the ReLU non-linearity in detail, we remark that we empirically also consider a range of other non-linearities including Logistic Sigmoid, SiLU, GELU, and Leaky-ReLU in SM B.1.

Equipped with the proposed technique for propagation, it is natural to assume that SDP may be computationally expensive; however, in the following section, we show how SDP can actually be very efficiently computed, even when propagating covariances.

### 3.3 COMPUTING THE SCALE PARAMETERS

For SDP, the location parameter of the output is simply the function value evaluation at the location of the input distribution. Therefore, we only need to give special consideration to the computation of the scale parameters. We provide pseudo-code and PyTorch implementations of SDP in SM D.

If we consider marginal Gaussian and marginal Cauchy distribution propagation, we need to carry one scale parameter for each location parameter through the network, i.e., each layer computes location and scales, which makes it around two times as expensive as regular propagation of a single point value. When comparing to marginal moment matching, marginal moment matching has a similar computational cost except for non-linear transformations being slightly more expensive if a respective marginal moment matching approximation exists. We remark that, for marginal moment matching, the locations (means) deviate from regular function evaluations and require the means to be matched.

An inherent limitation of any marginal propagation approach is that correlations between components of the random variables are discarded at each layer. For Gaussian distributions, this can lead to a layer-wise decay of scales, as illustrated, e.g., in the SM in Table 28 and Figure 10.

If we consider the non-marginal distribution case, we need to compute a scale matrix (e.g., the covariance matrix of a Gaussian). If we had to compute the scale matrix between any two layers of a neural network, it would become prohibitively expensive as the scale matrix for a layer with $n$ neurons has a size of $n \times n$. Because we are using a local linearization, we can compute the output scale matrix $\boldsymbol{\Sigma}_y$ of an $l$ layer network $f(x) = f_l(f_{l-1}(\cdots f_1(x) \cdots))$ as

$$\boldsymbol{\Sigma}_y = \boldsymbol{J}(f_l) \cdot \boldsymbol{J}(f_{l-1}) \cdots \boldsymbol{J}(f_1) \cdot \boldsymbol{\Sigma} \cdot \boldsymbol{J}(f_1)^\top \cdots \boldsymbol{J}(f_{l-1})^\top \cdot \boldsymbol{J}(f_l)^\top \tag{5}$$

where $\boldsymbol{J}(f_l)$ denotes the Jacobian of the $l$th layer. Via the chain rule, we can simplify the expression to

$$\boldsymbol{\Sigma}_y = \boldsymbol{J}(f) \cdot \boldsymbol{\Sigma} \cdot \boldsymbol{J}(f)^\top, \tag{6}$$

which means that we can, without approximation, compute the scale matrix based on $\boldsymbol{J}(f)$ (the Jacobian of the network.) Computing the Jacobian is relatively efficient at a cost of around $\min(k_{in}, k_{out})$ network evaluations where $k_{in}$ and $k_{out}$ correspond to the input and output dimensions, respectively. Exactly computing the moments for a full moment matching technique is intractable, even for very small networks. [33] proposed the approximate moment matching technique DVIA, which makes this idea tractable via approximation; however, their approach requires instantiating and propagating an $n \times n$ matrix through each layer, making DVIA expensive, particularly for large networks.

Finally, we would like to discuss how to evaluate the marginal Cauchy SDP without discarding any correlations within the propagation through the network. For this case, we can evaluate the Jacobian $\boldsymbol{J}(f)$ and derive the marginal scale of the output Cauchy distribution as $\boldsymbol{\gamma}_y = \boldsymbol{\gamma} \cdot \mathrm{Abs}(\boldsymbol{J}(f))^\top$.

### 3.4 EXPERIMENTAL EVALUATION OF SDP

To experimentally validate the performance of our approximation in the multi-dimensional multi-layer case, we train a set of neural networks with ReLU non-linearities trained with softmax cross-entropy on the Iris data set. We use Iris as it has small input and output dimensions, allowing us to use Monte Carlo as a feasible and accurate oracle estimator of the truth, and because we also compare to methods that would be too expensive for larger models. We compute the MC oracle with $10^6$ samples and evaluate it wrt. the intersection of probabilities / TV in Table 1 and wrt. the Wasserstein distance in Table 2. Each result is averaged over propagations through 10 independent models.

In Table 1, we can observe that SDP performs slightly superior to the more expensive DVIA approach. As another baseline, we estimate the Gaussian output distribution via propagating $k$ samples from the

Table 1: Evaluation of the accuracy of propagating Gaussians through a network with 4 ReLU layers trained on the Iris classification task [43]. We simulate the ground truth via an MC sampling approximation with $10^6$ samples as oracle and report the intersection of probability mass $(1 - TV)$.  †: [28], [20], [29], [18], [35], etc.

| $\sigma$ | SDP | DVIA [33] | MC Estimate $k{=}100$ | marginal SDP | mar. Moment M. (†) |
|---|---|---|---|---|---|
| 0.1 | **0.979** $\pm$ 0.020 | 0.972 $\pm$ 0.022 | 0.8081 $\pm$ 0.0877 | 0.236 $\pm$ 0.025 | **0.261** $\pm$ 0.025 |
| 1 | **0.874** $\pm$ 0.054 | 0.851 $\pm$ 0.054 | 0.7378 $\pm$ 0.0831 | **0.224** $\pm$ 0.023 | 0.219 $\pm$ 0.029 |
| 10 | **0.758** $\pm$ 0.040 | 0.703 $\pm$ 0.036 | 0.6735 $\pm$ 0.0626 | **0.218** $\pm$ 0.024 | 0.197 $\pm$ 0.017 |
| 100 | **0.687** $\pm$ 0.033 | 0.584 $\pm$ 0.047 | 0.6460 $\pm$ 0.0587 | **0.226** $\pm$ 0.028 | 0.172 $\pm$ 0.011 |
| 1000 | **0.680** $\pm$ 0.031 | 0.531 $\pm$ 0.051 | 0.6444 $\pm$ 0.0590 | **0.219** $\pm$ 0.024 | 0.170 $\pm$ 0.010 |

input distribution through the network, referred to as "MC Estimate" in Tab. 1. We can observe that SDP consistently outperforms MC estimation via 100 samples. Finally, when comparing marginal SDP to marginal moment matching, we can see that for input $\sigma = 0.1$ marginal moment matching has an advantage while marginal SDP performs better in the other cases. In Supplementary Material B.1, we present results for additional architectures (Tab. 7–14), numbers of MC samples (Tab. 15–18) and non-linearities, including evaluations for Leaky-ReLU (Tab. 23–26), logistic sigmoid activations (Tab. 27), SiLU activations [44] (Tab. 29–32), and GELU activations [45] (Tables 33–36).

Table 2: Evaluation of the accuracy of Gaussian SDP for a network with 4 ReLU layers (Iris). Here, we measure the $W_1$ Wasserstein distance between the ground truth distribution and our prediction, simulated via sets of $30\,000$ samples each. (Smaller is better.)

| $\sigma$ | SDP | DVIA | mar. Moment M.(†) |
|---|---|---|---|
| 0.01 | **0.0092** ± 0.0060 | **0.0092** ± 0.0059 | 0.1924 ± 0.1327 |
| 0.1 | **0.1480** ± 0.1574 | 0.1520 ± 0.1805 | 2.0809 ± 1.2755 |
| 1 | 7.7325 ± 5.1254 | **7.5794** ± 4.4295 | 20.9889 ± 7.7142 |

In Table 2, we compare the Wasserstein distances for SDP to DVIA and marginal moment matching. We observe that SDP is consistently competitive with the best results from DVIA, despite the substantially higher complexity of the latter.

In Fig. 3, we show the $W_1$ distance between the true distribution ReLU($X$) and SDP, as well as marginal moment matching. Here, $X \sim \mathcal{N}(\mu, \sigma = 0.1)$; and $\mu$ is given on the horizontal axis of the plot. We can see that SDP has smaller $W_1$ distances except for the area very close to $\mu = 0$. Additional architectures can be found in SM B.1 (Tab. 19–22).

In Tab. 3, we evaluate SDP for a CIFAR-10 ResNet-18 [46] model on $500$ images. We consider the TV between predicted distributions (distributions over classes) under Gaussian input noise on images. The ground truth is based on sets of $5\,000$ samples. We observe that SDP reduces the TV, illustrating the scalability of SDP to deeper and larger networks.

Figure 3: Plot of the log. of the $W_1$ distance between the true distribution ReLU($X$) (i.e., the distribution after a single ReLU) and SDP (blue) as well as marginal moment matching (orange).

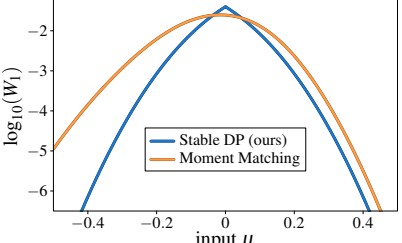

Table 3: Gaussian SDP for ResNet-18 on CIFAR-10. Displayed is the TV between the predictions under Gaussian input noise (smaller is better). For mar. MM, we use mar. SDP for MaxPool. Stds. in Tab. 37.

| $\sigma$ | SDP | mar. SDP | mar. MM |
|---|---|---|---|
| 0.01 | **0.00073** | 0.00308 | 0.02107 |
| 0.02 | **0.00324** | 0.00889 | 0.02174 |
| 0.05 | **0.01140** | 0.02164 | 0.02434 |
| 0.1 | **0.03668** | 0.04902 | 0.04351 |

## 3.5 LEARNING WITH STABLE DISTRIBUTION PROPAGATION

With the ability to propagate distributions through neural networks, the next step is to also use SDP for training. As the output of a neural network is a (multivariate) Gaussian or Cauchy distribution, the goal is to incorporate the covariances corresponding to the neural network's output into the loss.

For regression tasks, we minimize the negative log-likelihood, which is readily available and easy to optimize for Gaussian and Cauchy distributions including the multivariate (non-marginal) case. This enables the consideration of correlations between different output dimensions in the likelihood.

The more difficult case is classification with many classes. This is because the exact probability of a score among $n \gg 2$ scores being the maximum is intractable. Conventionally, in uncertainty agnostic training, this is resolved by assuming a Gumbel uncertainty in the output space, leading to the well-known softmax function / distribution. In particular, softmax corresponds to the probability of the argmax under isotropic Gumbel noise. However, for other distributions we are not aware of any closed-form solutions; for Gaussian distributions, the probability of argmax becomes hopelessly intractable, and finding closed-form correspondences may be generally considered an open problem. Previous work has proposed using marginal moment matching of softmax [20], Dirichlet outputs [20], and an approximation to the $(n-1)$-variate logistic distribution [18].

Instead of finding a surrogate that incorporates the marginal variances of the prediction, we propose to use the exact pair-wise probabilities for classification. To this end, we propose computing the pairwise probability of correct classification among all possible pairs in a one-vs-one setting, i.e., the true class score is compared to all false classes' scores and all $n - 1$ losses are averaged. This allows incorporating not only the scales but also the scale matrix (i.e., correlations) of a prediction.

The probability of pairwise correct classification for two RVs, $X$ (true class score) and $Y$ (false class score), is

$$\mathbb{P}(X > Y) = \mathbb{P}(X - Y > 0) = \int_0^\infty \mathrm{PDF}_{X-Y}(x)\,\mathrm{d}\,x = \mathrm{CDF}_{Y-X}(0)\ . \tag{7}$$

For a multivariate Gaussian distribution $(X, Y) \sim \mathcal{N}((\mu_X, \mu_Y), \boldsymbol{\Sigma})$ with covariance matrix $\Sigma$.

$$\mathbb{P}(X > Y) = \tfrac{1}{2}\left(1 + \mathrm{erf}\left(\frac{\mu_X - \mu_Y}{\sqrt{2(\sigma_{XX}^2 + \sigma_{YY}^2 - 2\sigma_{XY})}}\right)\right) \tag{8}$$

For Cauchy distributions $X \sim \mathcal{C}(x_X, \gamma_X)$ and $Y \sim \mathcal{C}(x_Y, \gamma_Y)$.

$$\mathbb{P}(X > Y) = \tfrac{1}{\pi}\left(\arctan\left(\frac{x_X - x_Y}{\gamma_X + \gamma_Y}\right)\right) + \tfrac{1}{2} \tag{9}$$

We refer to these losses as the Pairwise Distribution Losses for Gaussian resp. Cauchy distributions.

## 4    ADDITIONAL EXPERIMENTS

In this section, we evaluate the proposed method and loss functions on different tasks for distribution propagation. First, we investigate how well our method quantifies uncertainty on UCI [25] regression tasks. Second, we test how well our method can estimate model uncertainty for detection of out-of-distribution test data. In SM B.4, we investigate robustness against Gaussian and adversarial noise.

### 4.1    UNCERTAINTY QUANTIFICATION ON UCI DATA SETS

To evaluate uncertainty quantification on UCI regression tasks, we consider predicting calibrated intervals. We follow the experimental setting of Tagasovska *et al.* [9] and compare the proposed method to their Conditional Quantiles (SQR) method as well as PNNs, which was their best reported baseline. As SDP and PNNs are not mutually exclusive but are instead complementary, we consider applying our method to PNNs, i.e., we propagate distributions through the location component of the PNN and consider the uncertainty of the summation of both sources of uncertainty. Recall that SDP quantifies uncertainties arising from the input space and PNNs predict uncertainties in the output space, so we expect them to work well together at quantifying both types of data uncertainties. Details of the formulation for applying SDP to Gaussian PNNs can be found in Supplementary Material C.3.

We use the same model architecture, hyper-parameters, and evaluation metrics as [9] for all three methods. The evaluation metrics are Prediction Interval Coverage Probability (PICP), i.e., the fraction of test data points falling into the predicted intervals, and the Mean Prediction Interval Width (MPIW), i.e., the average width of the prediction intervals. In Tab. 4, following [9], we report the test PICP

Table 4: Results for the data uncertainty experiment on 8 UCI data sets. The task is to compute calibrated prediction intervals (PICP) while minimizing the Mean Prediction Interval Width (MPIW, in parentheses). Lower MPIW is better. Results averaged over 20 runs. MC Estimate is distribution propagation via sampling $k$ samples from the input distribution through the model. Results for MC Estimate with $k \in \{5, 1000\}$ are in Tab. 40.

| Data Set | SDP (ours) | SDP + PNN (ours) | PNN [9], [21], [34] |
|---|---|---|---|
| concrete | $0.92 \pm 0.03$ (**$0.25 \pm 0.02$**) | $0.94 \pm 0.02$ ($0.28 \pm 0.01$) | $0.94 \pm 0.03$ ($0.32 \pm 0.09$) |
| power | $0.94 \pm 0.01$ ($0.20 \pm 0.00$) | $0.95 \pm 0.01$ (**$0.15 \pm 0.01$**) | $0.94 \pm 0.01$ ($0.18 \pm 0.00$) |
| wine | $0.92 \pm 0.03$ ($0.45 \pm 0.03$) | $0.94 \pm 0.01$ (**$0.43 \pm 0.03$**) | $0.94 \pm 0.02$ ($0.49 \pm 0.03$) |
| yacht | $0.93 \pm 0.04$ ($0.06 \pm 0.01$) | $0.94 \pm 0.01$ (**$0.03 \pm 0.00$**) | $0.93 \pm 0.06$ (**$0.03 \pm 0.01$**) |
| naval | $0.94 \pm 0.02$ (**$0.02 \pm 0.00$**) | $0.95 \pm 0.02$ ($0.03 \pm 0.00$) | $0.96 \pm 0.01$ ($0.15 \pm 0.25$) |
| energy | $0.91 \pm 0.05$ (**$0.05 \pm 0.01$**) | $0.95 \pm 0.01$ ($0.10 \pm 0.01$) | $0.94 \pm 0.03$ ($0.12 \pm 0.18$) |
| boston | $0.93 \pm 0.04$ ($0.28 \pm 0.02$) | $0.94 \pm 0.03$ (**$0.25 \pm 0.04$**) | $0.94 \pm 0.03$ ($0.55 \pm 0.20$) |
| kin8nm | $0.95 \pm 0.01$ ($0.24 \pm 0.03$) | $0.94 \pm 0.01$ (**$0.19 \pm 0.01$**) | $0.93 \pm 0.01$ ($0.20 \pm 0.01$) |

| Data Set | MC Estimate ($k = 10$) | MC Estimate ($k = 100$) | SQR [9] |
|---|---|---|---|
| concrete | $0.93 \pm 0.04$ ($0.29 \pm 0.16$) | $0.95 \pm 0.03$ ($0.27 \pm 0.12$) | $0.94 \pm 0.03$ ($0.31 \pm 0.06$) |
| power | $0.93 \pm 0.08$ ($0.22 \pm 0.12$) | $0.94 \pm 0.05$ ($0.20 \pm 0.05$) | $0.93 \pm 0.01$ ($0.18 \pm 0.01$) |
| wine | $0.94 \pm 0.07$ ($0.51 \pm 0.14$) | $0.94 \pm 0.05$ ($0.45 \pm 0.06$) | $0.93 \pm 0.03$ ($0.45 \pm 0.04$) |
| yacht | $0.93 \pm 0.05$ ($0.15 \pm 0.12$) | $0.94 \pm 0.04$ ($0.09 \pm 0.04$) | $0.93 \pm 0.06$ ($0.06 \pm 0.04$) |
| naval | $0.93 \pm 0.06$ ($0.14 \pm 0.11$) | $0.94 \pm 0.04$ ($0.07 \pm 0.04$) | $0.95 \pm 0.02$ ($0.12 \pm 0.09$) |
| energy | $0.94 \pm 0.06$ ($0.12 \pm 0.10$) | $0.95 \pm 0.05$ ($0.08 \pm 0.06$) | $0.94 \pm 0.03$ ($0.08 \pm 0.03$) |
| boston | $0.94 \pm 0.05$ ($0.27 \pm 0.10$) | $0.95 \pm 0.04$ (**$0.25 \pm 0.08$**) | $0.92 \pm 0.06$ ($0.36 \pm 0.09$) |
| kin8nm | $0.93 \pm 0.06$ ($0.29 \pm 0.10$) | $0.94 \pm 0.03$ ($0.26 \pm 0.05$) | $0.93 \pm 0.01$ ($0.23 \pm 0.02$) |

and MPIW of those models where the validation PICP lies between $92.5\%$ and $97.5\%$ using the evaluation code provided by Tagasovska *et al.* [9]. The goal is to achieve a narrow interval (small MPIW) while keeping the optimal test PICP of $95\%$. As we quantify data uncertainty (and not model uncertainty) we note that the best deep ensemble [21] collapses to a single PNN; MC dropout [47] and Bayesian neural networks would focus on model and not on data uncertainty, see [9].

We can observe that SDP by itself performs competitively; compared to the baseline methods, which achieve predicting the narrowest intervals only on one of the data sets, SDP predicts the narrowest intervals on 3 data sets. Further, when applying SDP to PNNs, we improve over this result by achieving the narrowest intervals on 5 data sets. Notably, SDP + PNN does not only have the narrowest intervals but also the best overall prediction interval coverage. In each case, the best result (including on-par results) is either achieved by SDP or SDP + PNN and a general trend is that SDP + PNN leads to better calibrated prediction interval coverage. On 'boston', the stochastic MC Estimate approach with $k = 100$ propagations leads to the narrowest and best calibrated intervals; however, this result for MC Estimate comes at the substantially greater computational cost.

## 4.2 Out-of-Distribution Selective Prediction on MNIST and EMNIST

Selective prediction [48] is a formulation where instead of only predicting a class, a neural network can also abstain from a prediction if it is not certain. We benchmark selective prediction on MNIST [26] and use EMNIST letters [27] as out-of-distribution data. EMNIST letters is a data set that contains letters from A to Z in the same format as MNIST. We train a neural network on the MNIST training data set and then combine the MNIST test data set ($10\,000$ images) with $10\,000$ images from the EMNIST letter data set. This gives us a test data set of $20\,000$ images, $50\%$ of which are out-of-distribution samples and for which the model should abstain from prediction. We provide risk-coverage plots [48] of the selective prediction in Fig. 4 as well as the respective scores in Tab. 5. Risk-coverage plots report the empirical risk (i.e., the error) for each degree of coverage $\alpha$. That is, we select the $\alpha$ most certain predictions and report the error. This corresponds to a setting, where a predictor can abstain from prediction in $1-\alpha$ of the cases. We use risk-coverage area-under-the-curve

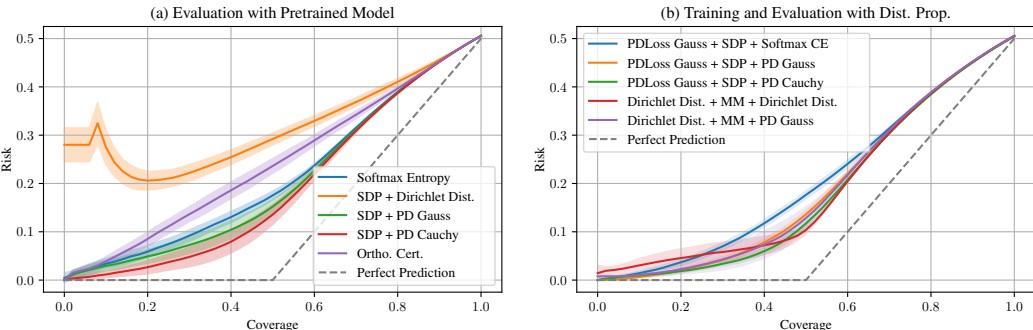

Figure 4: Selective prediction on MNIST with EMNIST letters as OOD data, evaluated on (a) pretrained off-the-shelf models as well as for (b) models trained with uncertainty propagation. Left: risk-coverage plots for off-the-shelf models trained with softmax cross-entropy. Right: models trained with uncertainty propagation. The grey line indicates perfect prediction. Results averaged over 10 runs.

Table 5: Selective prediction on MNIST with EMNIST letters as out-of-distribution data. We indicate the Risk-Coverage AUC (RCAUC) for (a) pretrained off-the-shelf models as well as for (b) models trained with uncertainty propagation (see also Fig. 4). GT baseline is $12.5\%$. Standard deviations are in Table 39.

| (a) Evaluation with Pretrained Model | | | |
|---|---|---|---|
| Train. Obj. | Dist. Prop. | Prediction | RCAUC |
| Softmax | — | Softmax Ent. | $21.4\%$ |
| Softmax | — | Ortho. Cert. | $24.2\%$ |
| Softmax | SDP | Dirichlet Dist. | $32.0\%$ |
| Softmax | SDP | PD Gauss | $20.4\%$ |
| Softmax | SDP | PD Cauchy | $\mathbf{19.2\%}$ |

| (b) Training and Evaluation with Dist. Prop. | | | |
|---|---|---|---|
| Train. Obj. | Dist. Prop. | Prediction | RCAUC |
| Dirichlet Dist. | mar. Moment M. | Dirichlet Dist. | $19.2\%$ |
| Dirichlet Dist. | mar. Moment M. | PD Gauss | $\mathbf{19.0\%}$ |
| PDLoss Gauss | SDP | Softmax Ent. | $20.6\%$ |
| PDLoss Gauss | SDP | PD Gauss | $19.0\%$ |
| PDLoss Gauss | SDP | PD Cauchy | $\mathbf{18.3\%}$ |

(AUC) to quantify the overall selective prediction performance. Smaller AUC implies that the network is more accurate on in-distribution test data while abstaining from making a wrong prediction on OOD examples. In this experiment, no correct predictions on OOD data are possible because the classifier can only predict numbers while the OOD data comprises letters.

In the first setting, in Figure 4 (a) and Table 5 (a), we train a neural network with conventional softmax cross-entropy to receive an existing off-the-shelf network. We compare five methods to compute certainty scores. First, we use the softmax entropy of the prediction, and apply Orthonormal Certificates from Tagasovska *et al.* [9]. Second, we propagate a distribution with $\sigma = 0.1$ through the network to obtain the covariances. Using these covariances, we use our pairwise Gaussian probabilities as well as the categorical probabilities of the Dirichlet outputs [20] to compute the entropies of the predictions. In the second setting, in Figure 4 (b) and Table 5 (b), we train the neural network using uncertainty propagation with the PD Gaussian loss as well as using marginal moment matching propagation and Dirichlet outputs [20]. We find that training with an uncertainty-aware training objective (pairwise Gaussian and Cauchy losses and Dirichlet outputs) improves the out-of-distribution detection in general. Further, we find that the pairwise Cauchy scores achieve the best out-of-distribution detection on a pre-trained model. Orthonormal certificates (OC) estimate null-space of data in the latent space to detect out-of-distribution examples. Digits and letters may share a similar null-space, making OC not as effective. The Dirichlet distribution does not perform well on a pretrained model because it is not designed for this scenario but instead designed to work in accordance with moment matching. We also observe that, for a model trained with Dirichlet outputs and marginal moment matching uncertainty propagation, selective prediction confidence scores of pairwise Gaussian and pairwise Cauchy offer an improvement in comparison to the Dirichlet output-based scores. This suggests that pairwise Cauchy scores are beneficial across various training approaches. The overall best accuracy (18.3% AUC) is achieved by training with an uncertainty-aware objective and evaluating using pairwise Cauchy. We also report RCAUC of a perfect predictor, i.e., one that always abstains on out-of-distribution data and always predicts correctly in-distribution.

## 4.3 RUNTIME ANALYSIS

In Table 6, we analyze the runtime for propagating distributions. While marginal distribution propagation causes only a slight overhead for small models, propagating distributions with full scale matrices is more expensive. Marginal moment matching (i.e., without scale matrix) is slightly more expensive than marginal SDP, as it requires evaluating additional

Table 6: Runtime Benchmark. Times per epoch on CIFAR-10 with a batch size of 128 on a single V100 GPU. DVIA was computationally too expensive in both settings.

| Model | regular | mar.MM. | mar. SDP | SDP |
|---|---|---|---|---|
| 3 Layer CNN | 6.29s | 6.33s | 6.31s | 20.8s |
| ResNet-18 | 9.85s | 30.7s* | 28.5s | 251s |

* as moment matching for some layers of ResNet (such as MaxPool for more than two inputs) is very expensive and does not have a closed-form solution in the literature, we use our approximation for these layers.

(scalar) functions at the non-linearities. The complexity factor (i.e., the factor of the cost of propagating a single point through the neural network) is linear in the minimum of input and output dimension for SDP, while it is constant for marginal propagation. As a reference, the exact propagation of the distributions (and not a parametric approximation as in this work) has an exponential runtime factor in the number of neurons [49]. The complexity factor of DVIA [33] is linear in the number of neurons in the largest layer, e.g., for the MNIST experiments in Section B.4, the computational time of a single forward pass with DVIA would be over 12 500 regular forward passes, which makes experiments exceeding Table 1 infeasible. See SM C for additional implementation details.

## 5 CONCLUSION

The application of deep learning across safety and reliability critical tasks emphasizes the need for reliable quantification of predictive uncertainties in deep learning. A major source of such uncertainty is data uncertainty, arising due to errors and limitations in the features and target measurements. In this work, we introduced SDP, an approach for propagating stable distributions through neural networks in a sampling-free and analytical manner. Our approach is superior to prior moment matching approaches in terms of accuracy and computational cost, while also enabling the propagation of a more diverse set of input distributions. Using SDP, we predict how input uncertainties lead to prediction uncertainties, and also introduced an extension for quantifying input and output uncertainties jointly. We demonstrated the utility of these new approaches across tasks involving prediction of calibrated confidence intervals and selective prediction of OOD data while comparing to other UQ approaches.

ACKNOWLEDGMENTS

This work was supported by the Goethe Center for Scientific Computing (G-CSC) at Goethe University Frankfurt, the IBM-MIT Watson AI Lab, the DFG in the Cluster of Excellence EXC 2117 "Centre for the Advanced Study of Collective Behaviour" (Project-ID 390829875), the Land Salzburg within the WISS 2025 project IDA-Lab (20102-F1901166-KZP and 20204-WISS/225/197-2019), the U.S. DOE Contract No. DE-AC02-76SF00515, and Zoox Inc.

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

# Supplementary Material for
# Uncertainty Quantification via Stable Distribution Propagation

## Table of Contents

## A   ADDITIONAL DISCUSSIONS AND PROOFS

### A.1   THEOREM 1

**Theorem 1.** *Local linearization provides the optimal Gaussian approximation of a univariate Gaussian distribution transformed by a ReLU non-linearity with respect to the total variation:*

$$\underset{(\tilde{\mu},\tilde{\sigma})}{\arg\min}\, TV(\boldsymbol{P},\boldsymbol{Q}) = \begin{cases} (\mu,\sigma) & \mu \geq 0 \\ (0,0) & \textit{otherwise} \end{cases} \tag{10}$$

$$\textit{where } \boldsymbol{P} = \mathcal{N}(\tilde{\mu},\tilde{\sigma}^2), \;\; \boldsymbol{Q} = \mathrm{ReLU}(\mathcal{N}(\mu,\sigma^2))$$

*Proof.* We distinguish 3 cases, $\mu < 0$, $\mu = 0$, and $\mu > 0$:

($\mu < 0$)     In the first case, the true distribution has a probability mass of $p_0 > 0.5$ at 0 because all values below 0 will be mapped to 0 and $\mathrm{CDF}(0) > 0.5$. As the approximation is parameterized as $(0,0)$, it has a probability mass of 1 at 0. Therefore, the TV is $|1 - p_0| < 0.5$. All parameterized distributions where $\sigma > 0$ have no probability mass at 0 and thus a TV of at least $|0 - p_0| = p_0 > 0.5$.

($\mu = 0$)     In this case, $\int_0^\infty |p - q|\, d\nu = 0$ because the distributions are equal on this domain. The true distribution has a probability mass of $p_0 = 0.5$ at 0, therefore the TV is $|0 - p_0| = 0.5$. In fact, all distributions with $\sigma > 0$ have a TV of at least 0.5 because $|0 - p_0| = 0.5$. The distribution parameterized by $(0,0)$ has the same TV at this point.

($\mu > 0$)     In this case, $p_0 = \mathrm{CDF}(0) < 0.5$. Further, again $\int_0^\infty |p - q|\, d\nu = 0$. Thus, the TV of our distribution is $|0 - p_0| = p_0 < 0.5$. All distributions with $\sigma > 0$ have a TV of at least $p_0$ because $|0 - p_0| = p_0$. The distribution parameterized by $(0,0)$ has a TV of $|1 - p_0| > 0.5$.

Thus, the approximation by linearization (Eq. 2) is the optimal approximation wrt. total variation. $\square$

**Corollary 2.** *Theorem 1 also applies to Cauchy distributions parameterized via $x_0, \gamma$ instead of $\mu, \sigma$.*

The proof of Theorem 1 also applies to Corollary 2. ($\mu$ is the median and $\sigma$ is the scale.)

## A.2 CONSIDERATIONS FOR $\alpha$-STABLE DISTRIBUTIONS

### A.2.1 AFFINE TRANSFORMATIONS FOR $\alpha$-STABLE DISTRIBUTIONS

Given a *marginal multivariate $\alpha$-stable distribution* $\boldsymbol{X} \sim \mathcal{S}(\boldsymbol{x}_0, \operatorname{diag}(\boldsymbol{\gamma}^2), \alpha, \beta)$ with $\boldsymbol{x}_0 \in \mathbb{R}^{1 \times n}, \boldsymbol{\gamma} \in \mathbb{R}_{\geq 0}^{1 \times n}$, the marginal Cauchy distribution of the transformation of $\boldsymbol{X}$ by a layer with a non-negative weight $\boldsymbol{A}_{\geq 0} \in \mathbb{R}_{\geq 0}^{m \times n}$ matrix and arbitrary bias $\boldsymbol{b}$ can be expressed via $\boldsymbol{x}_0 \mapsto \boldsymbol{x}_0 \boldsymbol{A}_{\geq 0}^\top + \boldsymbol{b}$ and $\boldsymbol{\gamma} \mapsto (\boldsymbol{\gamma}^\alpha \boldsymbol{A}_{\geq 0}^\top)^{1/\alpha}$. Here, $\alpha$ and $\beta$ remain invariant to the transformation.

Moreover, for networks with real-valued weights and for centered distributions ($\beta = 0$), we have $\boldsymbol{x}_0 \mapsto \boldsymbol{x}_0 \boldsymbol{A}^\top + \boldsymbol{b}$ and $\tilde{\boldsymbol{\gamma}} = (\boldsymbol{\gamma}^\alpha \operatorname{Abs}(\boldsymbol{A})^\top)^{1/\alpha}$ where $\tilde{\boldsymbol{\gamma}}$ is an upper bound for the transformation's output's scale vector whenever it exists and otherwise parameterizes a stable distribution with greater spread than the exact true distribution (which, in this case, does not lie in the family of stable distributions). This can be seen from addition being an upper bound for the spread under subtraction.

### A.2.2 THEOREM 1 FOR $\alpha$-STABLE DISTRIBUTIONS

While we only use Theorem 1 for Gaussian and Cauchy distributions in this paper, in the following we show that it also holds more generally for all stable distributions:

**Corollary 3.** *Theorem 1 also applies to any $\alpha$-stable distribution parameterized via a median $\mu$ and scale $c$ instead of $\sigma$. The (optional) skewness parameter $\beta$ remains the same if $\mu \geq 0$ and otherwise, when $c = 0$, the skewness parameter does not affect the distribution.*

The proof of Theorem 1 also applies to Corollary 3 provided that $\mu$ is the median (for skewed distributions, this differs from the location parameter; for symmetric distributions, this is equivalent to the location parameter).

## A.3 CONVOLUTIONS

For two-dimensional convolutions, we use the notation $\boldsymbol{Y} = \boldsymbol{X} * \mathbf{W} + \boldsymbol{B}$ where $\boldsymbol{X} \in \mathbb{R}^{c_0 \times n_0 \times n_1}$ is the input (image), $\mathbf{W} \in \mathbb{R}^{c_1 \times c_0 \times k_0 \times k_1}$ is the weight tensor, and $\boldsymbol{B}, \boldsymbol{Y} \in \mathbb{R}^{c_1 \times m_0 \times m_1}$ are the bias and output tensors with $c_0, c_1, k_0, k_1, n_0, n_1, m_0, m_1, m_0, m_1 \in \mathbb{N}_+$. Further, let $\boldsymbol{X}^{\mathsf{U}} \in \mathbb{R}^{m_0 \times m_1 \times c_0 \times k_0 \times k_1}$ be the unfolded sliding local blocks[2] from tensor $\boldsymbol{X}$ such that

$$\boldsymbol{Y}_{c_1 m_0 m_1} = \sum_{c_0 k_0 k_1} \boldsymbol{X}^{\mathsf{U}}_{m_0 m_1 c_0 k_0 k_1} \mathbf{W}_{c_1 c_0 k_0 k_1} + \boldsymbol{B} = \boldsymbol{X} * \mathbf{W} + \boldsymbol{B}. \tag{11}$$

*Multivariate Gaussian Distribution.* For the convolutional layer, $\boldsymbol{\mu} \in \mathbb{R}^{c_0 \times n_0 \times n_1}, \boldsymbol{\Sigma} \in \mathbb{R}^{c_0 \times n_0 \times n_1 \times c_0 \times n_0 \times n_1}$. Note that $\boldsymbol{\Sigma}^{\mathsf{U}} \in \mathbb{R}^{m_0 \times m_1 \times c_0 \times k_0 \times k_1 \times m_0 \times m_1 \times c_0 \times k_0 \times k_1}$ is the unfolded sliding local blocks covariance of the covariance tensor $\boldsymbol{\Sigma}$.

$$\boldsymbol{\mu} \mapsto \boldsymbol{\mu} * \mathbf{W} + \mathbf{B} \tag{12}$$

and

$$\boldsymbol{\Sigma}_{c_1 m_0 m_1 c_1' m_0' m_1'} \mapsto \sum_{c_0 k_0 k_1 c_0' k_0' k_1'} \mathbf{W}_{c_1 c_0 k_0 k_1} \boldsymbol{\Sigma}^{\mathsf{U}}_{m_0 m_1 c_0 k_0 k_1 m_0' m_1' c_0' k_0' k_1'} \mathbf{W}_{c_1' c_0' k_0' k_1'} \tag{13}$$

*Multivariate Marginal Gaussian Distribution.* For the convolutional layer, $\boldsymbol{\mu}, \boldsymbol{\sigma} \in \mathbb{R}^{c_0 \times n_0 \times n_1}$.

$$\boldsymbol{\mu} \mapsto \boldsymbol{\mu} * \mathbf{W} + \boldsymbol{B} \qquad \text{and} \qquad \boldsymbol{\sigma}^2 \mapsto \boldsymbol{\sigma}^2 * \mathbf{W}^2 \tag{14}$$

*Multivariate Marginal Cauchy Distribution.* For the convolutional layer, $\boldsymbol{x}_0, \boldsymbol{\gamma} \in \mathbb{R}^{c_0 \times n_0 \times n_1}$.

$$\boldsymbol{x}_0 \mapsto \boldsymbol{x}_0 * \mathbf{W} + \mathbf{B} \qquad \text{and} \qquad \boldsymbol{\gamma} \mapsto \boldsymbol{\gamma} * \operatorname{Abs}(\mathbf{W}) \tag{15}$$

---

[2]Einstein summation notation. The unfolded sliding local blocks are equivalent to `torch.unfold`.

# B  ADDITIONAL EXPERIMENTS

## B.1  PROPAGATING GAUSSIAN DISTRIBUTIONS

Tab. 7–36 show additional simulations with 1, 2, 4, and 6 hidden layers as well as ReLU, Leaky-ReLU, GELU, SiLU, and logistic sigmoid activations.

Tab. 28 considers the average ratio between predicted and true standard deviations.

As we did not find a moment matching method for GELU and SiLU in the literature, we omit moment matching in these cases. DVIA [33] is only applicable to ReLU among the non-linearities we consider. For marginal moment matching with logistic sigmoid, we used numerical integration as we are not aware of a closed-form solution.

Table 7: Accuracy of Gaussian SDP. Displayed is intersection of probabilities $(1-\mathrm{TV})$. The network is a **2** layer ReLU activated network with dimensions `4-100-3`, i.e., **1 ReLU** activation.

| $\sigma$ | SDP | DVIA | marginal SDP | marginal Moment M. |
|---|---|---|---|---|
| 0.1 | $0.9874 \pm 0.0079$ | $0.9875 \pm 0.0074$ | $0.4574 \pm 0.0275$ | $0.4552 \pm 0.0288$ |
| 1 | $0.9578 \pm 0.0188$ | $0.9619 \pm 0.0165$ | $0.4643 \pm 0.0235$ | $0.4439 \pm 0.0238$ |
| 10 | $0.8648 \pm 0.0206$ | $0.8982 \pm 0.0247$ | $0.4966 \pm 0.0232$ | $0.4580 \pm 0.0136$ |
| 100 | $0.8157 \pm 0.0231$ | $0.8608 \pm 0.0353$ | $0.5034 \pm 0.0248$ | $0.4640 \pm 0.0178$ |
| 1000 | $0.8103 \pm 0.0236$ | $0.8555 \pm 0.0372$ | $0.5041 \pm 0.0254$ | $0.4640 \pm 0.0182$ |

Table 8: Accuracy of Gaussian SDP. Displayed is intersection of probabilities $(1-\mathrm{TV})$. The network is a **3** layer ReLU activated network with dimensions `4-100-100-3`, i.e., **2 ReLU** activations.

| $\sigma$ | SDP | DVIA | marginal SDP | marginal Moment M. |
|---|---|---|---|---|
| 0.1 | $0.9861 \pm 0.0099$ | $0.9840 \pm 0.0112$ | $0.3070 \pm 0.0185$ | $0.3111 \pm 0.0193$ |
| 1 | $0.9259 \pm 0.0279$ | $0.9247 \pm 0.0270$ | $0.3123 \pm 0.0133$ | $0.3004 \pm 0.0136$ |
| 10 | $0.8093 \pm 0.0234$ | $0.8276 \pm 0.0255$ | $0.3463 \pm 0.0206$ | $0.2972 \pm 0.0170$ |
| 100 | $0.7439 \pm 0.0257$ | $0.8152 \pm 0.0347$ | $0.3931 \pm 0.0207$ | $0.3065 \pm 0.0237$ |
| 1000 | $0.7373 \pm 0.0259$ | $0.8061 \pm 0.0384$ | $0.3981 \pm 0.0188$ | $0.3079 \pm 0.0249$ |

Table 9: Accuracy of Gaussian SDP. Displayed is intersection of probabilities $(1-\mathrm{TV})$. The network is a **5** layer ReLU activated network with dimensions `4-100-100-100-100-3`, i.e., **4 ReLU** activations.

| $\sigma$ | SDP | DVIA | marginal SDP | marginal Moment M. |
|---|---|---|---|---|
| 0.1 | $0.9791 \pm 0.0202$ | $0.9720 \pm 0.0228$ | $0.2361 \pm 0.0250$ | $0.2611 \pm 0.0256$ |
| 1 | $0.8747 \pm 0.0546$ | $0.8519 \pm 0.0543$ | $0.2243 \pm 0.0237$ | $0.2195 \pm 0.0294$ |
| 10 | $0.7586 \pm 0.0407$ | $0.7035 \pm 0.0366$ | $0.2186 \pm 0.0244$ | $0.1976 \pm 0.0179$ |
| 100 | $0.6877 \pm 0.0333$ | $0.5845 \pm 0.0479$ | $0.2261 \pm 0.0282$ | $0.1724 \pm 0.0111$ |
| 1000 | $0.6808 \pm 0.0318$ | $0.5318 \pm 0.0516$ | $0.2193 \pm 0.0248$ | $0.1706 \pm 0.0109$ |

Table 10: Accuracy of Gaussian SDP. Displayed is intersection of probabilities $(1-\text{TV})$. The network is a **7** layer ReLU activated network with dimensions `4-100-100-100-100-100-100-3`, i.e., **6 ReLU** activations.

| $\sigma$ | SDP | DVIA | marginal SDP | marginal Moment M. |
|---|---|---|---|---|
| 0.1 | $0.9732 \pm 0.0219$ | $0.9660 \pm 0.0226$ | $0.2196 \pm 0.0208$ | $0.2601 \pm 0.0323$ |
| 1 | $0.8494 \pm 0.0853$ | $0.8166 \pm 0.0986$ | $0.2292 \pm 0.0303$ | $0.2236 \pm 0.0368$ |
| 10 | $0.7743 \pm 0.0477$ | $0.6309 \pm 0.0553$ | $0.2420 \pm 0.0295$ | $0.2327 \pm 0.0571$ |
| 100 | $0.7077 \pm 0.0348$ | $0.5265 \pm 0.0975$ | $0.3525 \pm 0.2337$ | $0.1800 \pm 0.0188$ |
| 1000 | $0.7013 \pm 0.0334$ | $0.5166 \pm 0.1223$ | $0.4422 \pm 0.2450$ | $0.1764 \pm 0.0175$ |

Table 11: Accuracy of Gaussian SDP. Displayed is intersection of probabilities $(1-\text{TV})$. The network is a **10** layer ReLU activated network with hidden dimension of 100, i.e., **9 ReLU** activations.

| $\sigma$ | SDP | DVIA | marginal SDP | marginal Moment M. |
|---|---|---|---|---|
| 0.1 | $0.9553 \pm 0.0555$ | $0.9294 \pm 0.0826$ | $0.2130 \pm 0.0330$ | $0.2492 \pm 0.0555$ |
| 1 | $0.8385 \pm 0.0725$ | $0.7796 \pm 0.0858$ | $0.3145 \pm 0.2083$ | $0.3050 \pm 0.2128$ |
| 10 | $0.7302 \pm 0.0637$ | $0.6083 \pm 0.0607$ | $0.3281 \pm 0.1879$ | $0.3446 \pm 0.2161$ |
| 100 | $0.6912 \pm 0.0536$ | $0.4522 \pm 0.1486$ | $0.3520 \pm 0.2270$ | $0.3009 \pm 0.2011$ |

Table 12: Accuracy of Gaussian SDP. Displayed is intersection of probabilities $(1-\text{TV})$. The network is a **15** layer ReLU activated network with hidden dimension of 100, i.e., **14 ReLU** activations.

| $\sigma$ | SDP | DVIA | marginal SDP | marginal Moment M. |
|---|---|---|---|---|
| 0.1 | $0.9276 \pm 0.0941$ | $0.8884 \pm 0.1473$ | $0.2183 \pm 0.0516$ | $0.2706 \pm 0.1145$ |
| 1 | $0.8356 \pm 0.0981$ | $0.7584 \pm 0.1354$ | $0.3032 \pm 0.1905$ | $0.3091 \pm 0.1942$ |
| 10 | $0.7068 \pm 0.0814$ | $0.5058 \pm 0.1059$ | $0.3019 \pm 0.1800$ | $0.4317 \pm 0.2573$ |
| 100 | $0.6600 \pm 0.0865$ | $0.3421 \pm 0.1477$ | $0.2617 \pm 0.1663$ | $0.4011 \pm 0.2463$ |

Table 13: Accuracy of Gaussian SDP. Displayed is intersection of probabilities $(1-\text{TV})$. The network is a **20** layer ReLU activated network with hidden dimension of 100, i.e., **19 ReLU** activations.

| $\sigma$ | SDP | DVIA | marginal SDP | marginal Moment M. |
|---|---|---|---|---|
| 0.1 | $0.9403 \pm 0.0844$ | $0.8787 \pm 0.1729$ | $0.2288 \pm 0.0414$ | $0.2791 \pm 0.1114$ |
| 1 | $0.8271 \pm 0.1501$ | $0.7119 \pm 0.1800$ | $0.3172 \pm 0.1742$ | $0.3090 \pm 0.1753$ |
| 10 | $0.6833 \pm 0.1437$ | $0.5029 \pm 0.1366$ | $0.3301 \pm 0.2178$ | $0.3633 \pm 0.2229$ |
| 100 | $0.6405 \pm 0.1402$ | $0.4669 \pm 0.1850$ | $0.4512 \pm 0.2563$ | $0.3821 \pm 0.2343$ |

Table 14: Accuracy of Gaussian SDP. Displayed is intersection of probabilities $(1-\text{TV})$. The network is a **25** layer ReLU activated network with hidden dimension of 100, i.e., **24 ReLU** activations.

| $\sigma$ | SDP | DVIA | marginal SDP | marginal Moment M. |
|---|---|---|---|---|
| 0.1 | $0.9244 \pm 0.1128$ | $0.8316 \pm 0.1826$ | $0.2287 \pm 0.0582$ | $0.2831 \pm 0.1093$ |
| 1 | $0.7592 \pm 0.1749$ | $0.6238 \pm 0.1798$ | $0.3374 \pm 0.1899$ | $0.3413 \pm 0.1867$ |
| 10 | $0.6157 \pm 0.1495$ | $0.4673 \pm 0.1637$ | $0.3496 \pm 0.2101$ | $0.3126 \pm 0.1934$ |
| 100 | $0.5556 \pm 0.1467$ | $0.4479 \pm 0.1310$ | $0.3977 \pm 0.1894$ | $0.2439 \pm 0.1515$ |

Table 15: Accuracy of Gaussian SDP like Table 7 but using MC Estimate.

| $\sigma$ | MC Estimate $k=5$ | MC Estimate $k=10$ | MC Estimate $k=100$ | MC Estimate $k=1000$ |
|---|---|---|---|---|
| 0.1 | $0.8067 \pm 0.0975$ | $0.8835 \pm 0.0562$ | $0.9642 \pm 0.0137$ | $0.9814 \pm 0.0101$ |
| 1 | $0.8051 \pm 0.0917$ | $0.8763 \pm 0.0508$ | $0.9397 \pm 0.0214$ | $0.9505 \pm 0.0205$ |
| 10 | $0.7902 \pm 0.0879$ | $0.8574 \pm 0.0461$ | $0.9097 \pm 0.0218$ | $0.9129 \pm 0.0224$ |
| 100 | $0.7660 \pm 0.0898$ | $0.8336 \pm 0.0438$ | $0.8802 \pm 0.0229$ | $0.8832 \pm 0.0245$ |
| 1000 | $0.7628 \pm 0.0897$ | $0.8306 \pm 0.0436$ | $0.8764 \pm 0.0232$ | $0.8794 \pm 0.0249$ |

Table 16: Accuracy of Gaussian SDP like Table 8 but using MC Estimate.

| $\sigma$ | MC Estimate $k=5$ | MC Estimate $k=10$ | MC Estimate $k=100$ | MC Estimate $k=1000$ |
|---|---|---|---|---|
| 0.1 | $0.7910 \pm 0.0948$ | $0.8752 \pm 0.0476$ | $0.9570 \pm 0.0183$ | $0.9730 \pm 0.0196$ |
| 1 | $0.7766 \pm 0.0843$ | $0.8473 \pm 0.0510$ | $0.8910 \pm 0.0342$ | $0.8903 \pm 0.0327$ |
| 10 | $0.7542 \pm 0.0797$ | $0.8066 \pm 0.0426$ | $0.8433 \pm 0.0291$ | $0.8424 \pm 0.0324$ |
| 100 | $0.7313 \pm 0.0750$ | $0.7817 \pm 0.0475$ | $0.8140 \pm 0.0319$ | $0.8120 \pm 0.0368$ |
| 1000 | $0.7289 \pm 0.0748$ | $0.7794 \pm 0.0477$ | $0.8112 \pm 0.0321$ | $0.8095 \pm 0.0370$ |

Table 17: Accuracy of Gaussian SDP like Table 9 but using MC Estimate.

| $\sigma$ | MC Estimate $k=5$ | MC Estimate $k=10$ | MC Estimate $k=100$ | MC Estimate $k=1000$ |
|---|---|---|---|---|
| 0.1 | $0.8081 \pm 0.0877$ | $0.8758 \pm 0.0510$ | $0.9451 \pm 0.0283$ | $0.9541 \pm 0.0324$ |
| 1 | $0.7378 \pm 0.0831$ | $0.7827 \pm 0.0811$ | $0.7933 \pm 0.0739$ | $0.7930 \pm 0.0808$ |
| 10 | $0.6735 \pm 0.0626$ | $0.6908 \pm 0.0373$ | $0.7142 \pm 0.0274$ | $0.7083 \pm 0.0265$ |
| 100 | $0.6460 \pm 0.0587$ | $0.6543 \pm 0.0444$ | $0.6691 \pm 0.0307$ | $0.6628 \pm 0.0330$ |
| 1000 | $0.6444 \pm 0.0590$ | $0.6518 \pm 0.0457$ | $0.6654 \pm 0.0316$ | $0.6586 \pm 0.0343$ |

Table 18: Accuracy of Gaussian SDP like Table 10 but using MC Estimate.

| $\sigma$ | MC Estimate $k=5$ | MC Estimate $k=10$ | MC Estimate $k=100$ | MC Estimate $k=1000$ |
|---|---|---|---|---|
| 0.1 | $0.8119 \pm 0.0875$ | $0.8783 \pm 0.0556$ | $0.9395 \pm 0.0316$ | $0.9447 \pm 0.0342$ |
| 1 | $0.7283 \pm 0.0972$ | $0.7561 \pm 0.1087$ | $0.7561 \pm 0.1028$ | $0.7501 \pm 0.1015$ |
| 10 | $0.6664 \pm 0.0627$ | $0.6840 \pm 0.0429$ | $0.6952 \pm 0.0293$ | $0.6904 \pm 0.0254$ |
| 100 | $0.6198 \pm 0.0538$ | $0.6285 \pm 0.0442$ | $0.6381 \pm 0.0302$ | $0.6331 \pm 0.0271$ |
| 1000 | $0.6171 \pm 0.0538$ | $0.6246 \pm 0.0453$ | $0.6344 \pm 0.0309$ | $0.6290 \pm 0.0277$ |

Table 19: Accuracy of Gaussian SDP for the **2** layer network from Table 7. Here, we measure the **Wasserstein** $W_1$ distance between the ground truth distribution and our prediction (smaller is better.)

| $\sigma$ | SDP | DVIA | marginal Moment M. |
|---|---|---|---|
| 0.01 | $0.0073 \pm 0.0012$ | $0.0073 \pm 0.0012$ | $0.0930 \pm 0.0218$ |
| 0.1 | $0.0823 \pm 0.0145$ | $0.0807 \pm 0.0144$ | $1.0587 \pm 0.2083$ |
| 1 | $1.6561 \pm 0.5361$ | $1.5195 \pm 0.5537$ | $10.5295 \pm 1.6149$ |

Table 20: Accuracy of Gaussian SDP for the **3** layer network from Table 8. Here, we measure the **Wasserstein** $W_1$ distance between the ground truth distribution and our prediction (smaller is better.)

| $\sigma$ | SDP | DVIA | marginal Moment M. |
|---|---|---|---|
| 0.01 | $0.0085 \pm 0.0025$ | $0.0085 \pm 0.0025$ | $0.1524 \pm 0.0506$ |
| 0.1 | $0.1017 \pm 0.0351$ | $0.0998 \pm 0.0350$ | $1.7041 \pm 0.4962$ |
| 1 | $3.5549 \pm 1.4615$ | $3.3172 \pm 1.2598$ | $16.8069 \pm 3.2651$ |

Table 21: Accuracy of Gaussian SDP for the **5** layer network from Table 9. Here, we measure the **Wasserstein** $W_1$ distance between the ground truth distribution and our prediction (smaller is better.)

| $\sigma$ | SDP | DVIA | marginal Moment M. |
|---|---|---|---|
| 0.01 | $0.0093 \pm 0.0046$ | $0.0092 \pm 0.0046$ | $0.1956 \pm 0.1072$ |
| 0.1 | $0.1343 \pm 0.0778$ | $0.1322 \pm 0.0830$ | $2.1402 \pm 1.0347$ |
| 1 | $6.6188 \pm 3.9645$ | $6.4090 \pm 3.2720$ | $21.4352 \pm 5.8368$ |

Table 22: Accuracy of Gaussian SDP for the **7** layer network from Table 10. Here, we measure the **Wasserstein** $W_1$ distance between the ground truth distribution and our prediction (smaller is better.)

| $\sigma$ | SDP | DVIA | marginal Moment M. |
|---|---|---|---|
| 0.01 | $0.0092 \pm 0.0060$ | $0.0092 \pm 0.0059$ | $0.1924 \pm 0.1327$ |
| 0.1 | $0.1480 \pm 0.1574$ | $0.1520 \pm 0.1805$ | $2.0809 \pm 1.2755$ |
| 1 | $7.7325 \pm 5.1254$ | $7.5794 \pm 4.4295$ | $20.9889 \pm 7.7142$ |

Table 23: Accuracy of Gaussian SDP. Displayed is intersection of probabilities $(1-\text{TV})$. The network is a **2** layer Leaky-ReLU activated network with dimensions `4-100-3`, i.e., **1 Leaky-ReLU** activation with negative slope $\alpha = 0.1$.

| $\sigma$ | SDP | marginal SDP | marginal Moment M. |
|---|---|---|---|
| 0.1 | $0.9852 \pm 0.0065$ | $0.4484 \pm 0.0204$ | $0.4481 \pm 0.0208$ |
| 1 | $0.9624 \pm 0.0152$ | $0.4547 \pm 0.0183$ | $0.4417 \pm 0.0172$ |
| 10 | $0.8876 \pm 0.0194$ | $0.4838 \pm 0.0198$ | $0.4454 \pm 0.0118$ |
| 100 | $0.8458 \pm 0.0226$ | $0.4912 \pm 0.0203$ | $0.4581 \pm 0.0139$ |
| 1000 | $0.8411 \pm 0.0232$ | $0.4918 \pm 0.0207$ | $0.4594 \pm 0.0141$ |

Table 24: Accuracy of Gaussian SDP. Displayed is intersection of probabilities $(1-\text{TV})$. The network is a **3** layer Leaky-ReLU activated network with dimensions `4-100-100-3`, i.e., **2 Leaky-ReLU** activations with negative slope $\alpha = 0.1$.

| $\sigma$ | SDP | marginal SDP | marginal Moment M. |
|---|---|---|---|
| 0.1 | $0.9806 \pm 0.0085$ | $0.3020 \pm 0.0162$ | $0.3063 \pm 0.0171$ |
| 1 | $0.9324 \pm 0.0255$ | $0.3055 \pm 0.0129$ | $0.3007 \pm 0.0124$ |
| 10 | $0.8321 \pm 0.0254$ | $0.3343 \pm 0.0180$ | $0.2963 \pm 0.0147$ |
| 100 | $0.7716 \pm 0.0294$ | $0.3769 \pm 0.0203$ | $0.3040 \pm 0.0189$ |
| 1000 | $0.7651 \pm 0.0297$ | $0.3825 \pm 0.0190$ | $0.3056 \pm 0.0198$ |

Table 25: Accuracy of Gaussian SDP. Displayed is intersection of probabilities $(1-\mathrm{TV})$. The network is a **5** layer Leaky-ReLU activated network with dimensions `4-100-100-100-100-3`, i.e., **4 Leaky-ReLU** activations with negative slope $\alpha = 0.1$.

| $\sigma$ | SDP | marginal SDP | marginal Moment M. |
|---|---|---|---|
| 0.1 | $0.9689 \pm 0.0163$ | $0.2325 \pm 0.0224$ | $0.2566 \pm 0.0249$ |
| 1 | $0.8856 \pm 0.0460$ | $0.2260 \pm 0.0197$ | $0.2218 \pm 0.0236$ |
| 10 | $0.7638 \pm 0.0393$ | $0.2146 \pm 0.0220$ | $0.1975 \pm 0.0169$ |
| 100 | $0.6912 \pm 0.0352$ | $0.2364 \pm 0.0227$ | $0.1756 \pm 0.0112$ |
| 1000 | $0.6835 \pm 0.0337$ | $0.2361 \pm 0.0216$ | $0.1743 \pm 0.0116$ |

Table 26: Accuracy of Gaussian SDP. Displayed is intersection of probabilities $(1-\mathrm{TV})$. The network is a **7** layer Leaky-ReLU activated network with dimensions `4-100-100-100-100-100-100-3`, i.e., **6 Leaky-ReLU** activations with negative slope $\alpha = 0.1$.

| $\sigma$ | SDP | marginal SDP | marginal Moment M. |
|---|---|---|---|
| 0.1 | $0.9585 \pm 0.0203$ | $0.2197 \pm 0.0199$ | $0.2519 \pm 0.0292$ |
| 1 | $0.8573 \pm 0.0720$ | $0.2251 \pm 0.0289$ | $0.2220 \pm 0.0351$ |
| 10 | $0.7701 \pm 0.0465$ | $0.2332 \pm 0.0301$ | $0.2243 \pm 0.0259$ |
| 100 | $0.7047 \pm 0.0378$ | $0.2940 \pm 0.1864$ | $0.1779 \pm 0.0171$ |
| 1000 | $0.6979 \pm 0.0371$ | $0.3518 \pm 0.2209$ | $0.1746 \pm 0.0159$ |

Table 27: Accuracy of Gaussian SDP. Displayed is intersection of probabilities $(1-\mathrm{TV})$. The network is a **5** layer Logistic Sigmoid activated network with dimensions `4-100-100-100-100-3`, i.e., **4 Logistic Sigmoid** activations.

| $\sigma$ | SDP | marginal SDP | marginal Moment M. |
|---|---|---|---|
| 0.1 | 0.9890 | 0.7809 | 0.7793 |
| 1 | 0.9562 | 0.7880 | 0.7912 |
| 10 | 0.8647 | 0.8656 | 0.7674 |
| 100 | 0.8443 | 0.8442 | 0.8027 |
| 1000 | 0.8440 | 0.8440 | 0.8070 |

Table 28: Gaussian SDP output std ratios (value of 1 is optimal). The network is a **5** layer ReLU activated network with dimensions `4-100-100-100-100-3`, i.e., **4 ReLU** activations. Displayed is the average ratio between the output standard deviations. The 3 values, correspond to the three output dimensions of the model. We find that both our method with covariances as well as DVIA achieve a good accuracy in this setting, while the methods which do not consider covariances (ours (w/o cov.) and Moment Matching) underestimate the output standard deviations by a large factor. For small input standard deviations, our method w/ cov. as well as DVIA perform better. For large input standard deviations, our method tends to overestimate the output standard deviation, while DVIA underestimates the standard deviation. The ratios for our method and DVIA (while going into opposite directions) are similar, e.g., 2.6095 for our method and $1/0.3375 = 2.9629$ for DVIA.

| $\sigma$ | SDP | | | DVIA | | | marginal SDP | | | marginal Moment M. | | |
|---|---|---|---|---|---|---|---|---|---|---|---|---|
| 0.1 | 0.9933 | 1.0043 | 0.9954 | 0.9904 | 0.9917 | 0.9906 | 0.0126 | 0.0180 | 0.0112 | 0.0255 | 0.0373 | 0.0211 |
| 1 | 0.9248 | 1.0184 | 0.9598 | 0.8462 | 0.9256 | 0.8739 | 0.0103 | 0.0170 | 0.0100 | 0.0101 | 0.0166 | 0.0097 |
| 10 | 1.9816 | 2.2526 | 2.0629 | 0.6035 | 0.6664 | 0.6172 | 0.0207 | 0.0359 | 0.0205 | 0.0166 | 0.0269 | 0.0157 |
| 100 | 2.5404 | 2.8161 | 2.6217 | 0.3721 | 0.4194 | 0.3849 | 0.0264 | 0.0450 | 0.0260 | 0.0186 | 0.0296 | 0.0174 |
| 1000 | 2.6095 | 2.8857 | 2.6906 | 0.3375 | 0.3839 | 0.3503 | 0.0271 | 0.0462 | 0.0267 | 0.0188 | 0.0298 | 0.0175 |

Table 29: Accuracy of Gaussian SDP. The network is a **2** layer SiLU activated network with dimensions `4-100-3`, i.e., **1 SiLU** activation. Displayed is $1-\mathrm{TV}$.

| $\sigma$ | SDP | marginal SDP |
|---|---|---|
| 0.1 | $0.9910 \pm 0.0025$ | $0.4494 \pm 0.0192$ |
| 1 | $0.9813 \pm 0.0076$ | $0.4550 \pm 0.0158$ |
| 10 | $0.8635 \pm 0.0180$ | $0.5158 \pm 0.0251$ |
| 100 | $0.8104 \pm 0.0201$ | $0.5328 \pm 0.0282$ |
| 1000 | $0.8050 \pm 0.0205$ | $0.5347 \pm 0.0288$ |

Table 33: Accuracy of Gaussian SDP. The network is a **2** layer GELU activated network with dimensions `4-100-3`, i.e., **1 GELU** activation. Displayed is $1-\mathrm{TV}$.

| $\sigma$ | SDP | marginal SDP |
|---|---|---|
| 0.1 | $0.9875 \pm 0.0075$ | $0.4567 \pm 0.0317$ |
| 1 | $0.9732 \pm 0.0096$ | $0.4682 \pm 0.0225$ |
| 10 | $0.8530 \pm 0.0185$ | $0.5204 \pm 0.0266$ |
| 100 | $0.8039 \pm 0.0211$ | $0.5251 \pm 0.0278$ |
| 1000 | $0.7987 \pm 0.0216$ | $0.5255 \pm 0.0282$ |

Table 30: Accuracy of Gaussian SDP. The network is a **3** layer SiLU activated network with dimensions `4-100-100-3`, i.e., **2 SiLU** activations. Displayed is $1-\mathrm{TV}$.

| $\sigma$ | SDP | marginal SDP |
|---|---|---|
| 0.1 | $0.9876 \pm 0.0051$ | $0.3056 \pm 0.0178$ |
| 1 | $0.9658 \pm 0.0121$ | $0.3076 \pm 0.0122$ |
| 10 | $0.8067 \pm 0.0136$ | $0.3531 \pm 0.0189$ |
| 100 | $0.7425 \pm 0.0162$ | $0.3806 \pm 0.0261$ |
| 1000 | $0.7365 \pm 0.0167$ | $0.3829 \pm 0.0233$ |

Table 34: Accuracy of Gaussian SDP. The network is a **3** layer GELU activated network with dimensions `4-100-100-3`, i.e., **2 GELU** activations. Displayed is $1-\mathrm{TV}$.

| $\sigma$ | SDP | marginal SDP |
|---|---|---|
| 0.1 | $0.9860 \pm 0.0054$ | $0.3065 \pm 0.0244$ |
| 1 | $0.9453 \pm 0.0169$ | $0.3129 \pm 0.0164$ |
| 10 | $0.8024 \pm 0.0128$ | $0.3560 \pm 0.0176$ |
| 100 | $0.7446 \pm 0.0155$ | $0.3966 \pm 0.0237$ |
| 1000 | $0.7390 \pm 0.0159$ | $0.4004 \pm 0.0222$ |

Table 31: Accuracy of Gaussian SDP. The network is a **5** layer SiLU activated network with dimensions `4-100-100-100-100-3`, i.e., **4 SiLU** activations. Displayed is $1-\mathrm{TV}$.

| $\sigma$ | SDP | marginal SDP |
|---|---|---|
| 0.1 | $0.9704 \pm 0.0126$ | $0.2471 \pm 0.0277$ |
| 1 | $0.9174 \pm 0.0343$ | $0.2414 \pm 0.0096$ |
| 10 | $0.7478 \pm 0.0342$ | $0.2357 \pm 0.0141$ |
| 100 | $0.6846 \pm 0.0241$ | $0.2453 \pm 0.0254$ |
| 1000 | $0.6789 \pm 0.0238$ | $0.2371 \pm 0.0236$ |

Table 35: Accuracy of Gaussian SDP. The network is a **5** layer GELU activated network with dimensions `4-100-100-100-100-3`, i.e., **4 GELU** activations. Displayed is $1-\mathrm{TV}$.

| $\sigma$ | SDP | marginal SDP |
|---|---|---|
| 0.1 | $0.9706 \pm 0.0104$ | $0.2389 \pm 0.0287$ |
| 1 | $0.8916 \pm 0.0359$ | $0.2442 \pm 0.0131$ |
| 10 | $0.7607 \pm 0.0247$ | $0.2460 \pm 0.0220$ |
| 100 | $0.6975 \pm 0.0184$ | $0.2532 \pm 0.0295$ |
| 1000 | $0.6915 \pm 0.0193$ | $0.2465 \pm 0.0288$ |

Table 32: Accuracy of Gaussian SDP. The network is a **7** layer SiLU activated network with dimensions `4-100-100-100-100-100-100-3`, i.e., **6 SiLU** activations. Displayed is $1-\mathrm{TV}$.

| $\sigma$ | SDP | marginal SDP |
|---|---|---|
| 0.1 | $0.9147 \pm 0.0421$ | $0.2326 \pm 0.0345$ |
| 1 | $0.8248 \pm 0.0777$ | $0.2348 \pm 0.1062$ |
| 10 | $0.7293 \pm 0.0388$ | $0.2762 \pm 0.1315$ |
| 100 | $0.6726 \pm 0.0411$ | $0.2326 \pm 0.0282$ |
| 1000 | $0.6675 \pm 0.0417$ | $0.2363 \pm 0.0309$ |

Table 36: Accuracy of Gaussian SDP. The network is a **7** layer GELU activated network with dimensions `4-100-100-100-100-100-100-3`, i.e., **6 GELU** activations. Displayed is $1-\mathrm{TV}$.

| $\sigma$ | SDP | marginal SDP |
|---|---|---|
| 0.1 | $0.9262 \pm 0.0294$ | $0.2261 \pm 0.0352$ |
| 1 | $0.8387 \pm 0.0858$ | $0.2710 \pm 0.1547$ |
| 10 | $0.7342 \pm 0.0266$ | $0.3557 \pm 0.2025$ |
| 100 | $0.6898 \pm 0.0218$ | $0.2244 \pm 0.0259$ |
| 1000 | $0.6854 \pm 0.0220$ | $0.2235 \pm 0.0264$ |

## B.2 Additional Standard Deviations for Results in the Main Paper

Table 37: Gaussian SDP for **ResNet-18** on CIFAR-10. Displayed is the total variation distance (TV) between the distribution of predicted classes under Gaussian input noise on images (oracle via 5 000 samples), and the distribution of classes predicted using respective distribution propagation methods. Evaluated on 500 test images. For marginal moment matching, we use marginal SDP for the MaxPool operation as there is no respective formulation. Here, smaller values are better. We remark that the standard deviations are between the TVs of individual propagations (different images).

| $\sigma$ | SDP | marginal SDP | marginal Moment M. |
|---|---|---|---|
| 0.01 | **0.00073** $\pm$ 0.00703 | 0.00308 $\pm$ 0.02847 | 0.02107 $\pm$ 0.13415 |
| 0.02 | **0.00324** $\pm$ 0.02346 | 0.00889 $\pm$ 0.05239 | 0.02174 $\pm$ 0.12188 |
| 0.05 | **0.01140** $\pm$ 0.05879 | 0.02164 $\pm$ 0.10290 | 0.02434 $\pm$ 0.11390 |
| 0.1 | **0.03668** $\pm$ 0.11126 | 0.04902 $\pm$ 0.15999 | 0.04351 $\pm$ 0.14315 |

Table 38: Additional results corresponding to Table 4 reporting the test **negative log likelihood** (NLL).

| Data Set | SDP (ours) | SDP + PNN (ours) | PNN [9] |
|---|---|---|---|
| concrete | $-0.816 \pm 0.088$ | $\mathbf{-0.898} \pm 0.063$ | $-0.774 \pm 0.125$ |
| power | $-0.838 \pm 0.048$ | $\mathbf{-0.979} \pm 0.044$ | $-0.919 \pm 0.037$ |
| wine | $0.266 \pm 0.136$ | $\mathbf{0.197} \pm 0.062$ | $0.226 \pm 0.103$ |
| yacht | $-2.543 \pm 0.366$ | $\mathbf{-3.326} \pm 0.212$ | $-3.015 \pm 0.304$ |
| naval | $\mathbf{-2.127} \pm 0.032$ | $-1.691 \pm 0.061$ | $-1.544 \pm 0.416$ |
| energy | $\mathbf{-1.135} \pm 0.014$ | $-0.998 \pm 0.021$ | $-0.875 \pm 0.342$ |
| boston | $-0.966 \pm 0.045$ | $\mathbf{-1.240} \pm 0.072$ | $-0.574 \pm 0.165$ |
| kin8nm | $-0.198 \pm 0.121$ | $\mathbf{-0.293} \pm 0.151$ | $-0.255 \pm 0.166$ |

| Data Set | MC E. ($k$=10) | MC E. ($k$=100) | SQR [9] |
|---|---|---|---|
| concrete | $-0.749 \pm 0.159$ | $-0.721 \pm 0.113$ | $-0.761 \pm 0.089$ |
| power | $-0.788 \pm 0.232$ | $-0.864 \pm 0.137$ | $-0.903 \pm 0.067$ |
| wine | $0.231 \pm 0.223$ | $0.211 \pm 0.062$ | $0.218 \pm 0.134$ |
| yacht | $-1.742 \pm 0.769$ | $-2.716 \pm 0.261$ | $-2.885 \pm 0.436$ |
| naval | $-0.887 \pm 0.661$ | $-1.631 \pm 0.051$ | $-1.611 \pm 0.232$ |
| energy | $-0.412 \pm 0.611$ | $-0.977 \pm 0.040$ | $-0.920 \pm 0.051$ |
| boston | $-0.688 \pm 0.274$ | $-0.831 \pm 0.052$ | $-0.632 \pm 0.121$ |
| kin8nm | $-0.029 \pm 0.254$ | $-0.237 \pm 0.066$ | $-0.221 \pm 0.147$ |

Table 39: Extension of Table 5 including standard deviations.

(a) Evaluation with Pretrained Model

| Train. Obj. | Dist. Prop. | Prediction | RCAUC |
|---|---|---|---|
| Softmax | — | Softmax Ent. | $21.4\% \pm 0.6\%$ |
| Softmax | — | Ortho. Cert. | $24.2\% \pm 0.9\%$ |
| Softmax | SDP | Dirichlet Dist. | $32.0\% \pm 1.2\%$ |
| Softmax | SDP | PD Gauss | $20.4\% \pm 0.9\%$ |
| Softmax | SDP | PD Cauchy | $\mathbf{19.2\%} \pm 1.0\%$ |

(b) Training and Evaluation with Dist. Prop.

| Train. Obj. | Dist. Prop. | Prediction | RCAUC |
|---|---|---|---|
| Dirichlet Dist. | mar. Moment M. | Dirichlet Dist. | $19.2\% \pm 0.8\%$ |
| Dirichlet Dist. | mar. Moment M. | PD Gauss | $\mathbf{19.0\%} \pm 1.0\%$ |
| PDLoss Gauss | SDP | Softmax Ent. | $20.6\% \pm 0.4\%$ |
| PDLoss Gauss | SDP | PD Gauss | $19.0\% \pm 0.3\%$ |
| PDLoss Gauss | SDP | PD Cauchy | $\mathbf{18.3\%} \pm 0.3\%$ |

## B.3 ADDITIONAL RESULTS FOR UCI UQ WITH MC ESTIMATE

Table 40 shows additional results for the UCI data set uncertainty quantification regression experiment. Here, we consider $k \in \{5, 1000\}$. $k = 5$ frequently fails maintaining adequate prediction interval coverage probability (PICP). On the other hand $k = 1000$ is computationally quite expensive due to the requirement of 1000 neural network evaluations per data point.

Table 40: Additional results corresponding to Table 4 for $k \in \{5, 1000\}$ samples.

| Data Set | MC Estimate ($k = 5$) | MC Estimate ($k = 1000$) |
|---|---|---|
| concrete | $0.91 \pm 0.06 \ (0.31 \pm 0.20)$ | $0.95 \pm 0.03 \ (0.26 \pm 0.09)$ |
| power | $0.87 \pm 0.12 \ (0.26 \pm 0.15)$ | $0.94 \pm 0.04 \ (0.20 \pm 0.05)$ |
| wine | $0.92 \pm 0.08 \ (0.66 \pm 0.25)$ | $0.94 \pm 0.05 \ (0.44 \pm 0.05)$ |
| yacht | $0.93 \pm 0.06 \ (0.32 \pm 0.15)$ | $0.94 \pm 0.04 \ (0.08 \pm 0.03)$ |
| naval | $0.92 \pm 0.08 \ (0.22 \pm 0.20)$ | $0.94 \pm 0.03 \ (0.05 \pm 0.02)$ |
| energy | $0.92 \pm 0.07 \ (0.22 \pm 0.20)$ | $0.95 \pm 0.04 \ (0.07 \pm 0.05)$ |
| boston | $0.93 \pm 0.06 \ (0.33 \pm 0.15)$ | $0.95 \pm 0.04 \ (0.25 \pm 0.08)$ |
| kin8nm | $0.91 \pm 0.08 \ (0.34 \pm 0.14)$ | $0.94 \pm 0.02 \ (0.24 \pm 0.04)$ |

## B.4 GAUSSIAN AND ADVERSARIAL NOISE ROBUSTNESS

The robustness of neural networks has received significant attention in the past years [50]–[52]. There are many methods for robustifying neural networks against random noise in the inputs or adversarially crafted perturbations. While our method is not explicitly designed for these purposes, however, it can be observed that it achieves meaningful robustness improvements. We evaluate the robustness of classification models trained with uncertainty propagation on the MNIST [26] and CIFAR-10 [53] data sets. We consider the following training scenarios: First, we propagate normal distributions with and without covariance and train using the pairwise Gaussian loss. Second, we train with Cauchy distribution propagation using the pairwise Cauchy loss. As for baselines, we compare to training with softmax cross-entropy as well as moment matching propagation with the Dirichlet output loss [20]. The network architectures, hyper-parameters, as well as the results on CIFAR-10 are presented in Sections B.4.1 and C.

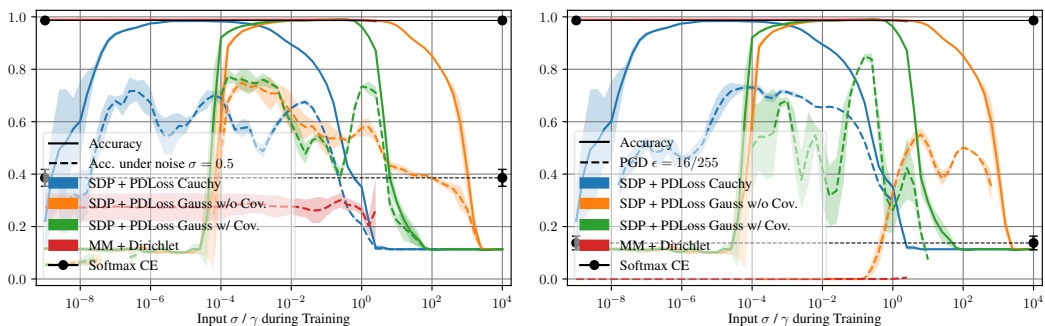

Figure 5: Robustness of CNNs on the MNIST data set against Gaussian noise (left) as well as PGD adversarial noise (right). The continuous lines show the test accuracy and the dashed lines show the robust accuracy. The black lines indicate the softmax cross-entropy trained baseline. Results are averaged over 3 runs. Results on CIFAR-10 and further settings are in Supplementary Material B.4.1.

**Random Gaussian Noise** To evaluate the random noise robustness of models trained with uncertainty propagation, we add random Gaussian per-pixel noise during evaluation. Specifically, we use input noise with $\sigma = 0.5$ (see Supplementary Material B.4.1 for $\sigma \in \{0.25, 0.75\}$) and clamp the image to the original pixel value range (between 0 and 1). Figure 5 (left) shows the results for input standard deviations/scales from $10^{-9}$ to $10^{4}$. For every input standard deviation, we train separate models. For each method, the range of good hyperparameters is around $4 - 6$ orders of magnitude. We observe when training with uncertainty propagation and the pairwise Gaussian/Cauchy losses, the models are substantially more robust than with conventional training on the MNIST data set.

**Adversarial Noise**  To evaluate the adversarial robustness of models trained with uncertainty propagation, we use the projected gradient descent (PGD) attack [51] as a common benchmark. For measuring the adversarial robustness, we use $L_\infty$-bounded PGD, for MNIST with $\epsilon = 16/255$ (and with $\epsilon = 8/255$ in the SM), and for CIFAR with $\epsilon \in \{3/255, 4/255\}$. Figure 5 (right) shows the adversarial robustness of CNNs on distributions. We can see that the proposed method is competitive with the accuracy of the softmax cross-entropy loss and can even outperform it. It further shows that the proposed method, in combination with pairwise probability classification losses, outperforms both baselines in each case. We find that the adversarially most robust training method is propagating normal distributions with covariances, with a robust accuracy of $85\%$ with $\epsilon = 16/255$ on the MNIST data set compared to a robust accuracy of $14\%$ for models trained with softmax cross-entropy, while the Dirichlet loss with moment-matching offers no robustness gains in this experiment. When training with SDP Gauss w/o cov., the models are only robust against adversarial noise for rather large input standard deviations. This effect is inverse for random Gaussian noise, where robustness against random noise is achieved when input standard deviations are rather small. On our models, we perform PGD with the PDLoss as well as PGD with softmax cross-entropy to ensure a thorough evaluation. In Section B.4.1, we provide results where a softmax CE-based objective is used by the adversary: our models are more robust against the CE-based attack than against the PDLoss-based attacks which are reported in the plot. While the proposed techniques are not as robust as methods specifically designed for that purpose, e.g., adversarial training [51], they show substantial improvements over training with cross-entropy.

### B.4.1 Additional Plots

In Fig. 7, we present an extension to Fig. 5 which demonstrates random and adversarial noise robustness for alternative noise intensities.

In Fig. 8 and 9, we present an additional demonstration of the adversarial and random noise robustness of CNNs on distributions. Note that we do not include results for normal distributions with covariances (as in Figures 5 and 7). Here, the emphasis is set on using both MNIST and CIFAR.

In Fig. 6, we show the robustness where a cross-entropy based objective (green) is used by the adversary. A cross-entropy based objective could be used in practice in multiple scenarios: For example, publishing just the weights of the model without information about the objective function, or by using a standard library for adversarial attacks that does not require or support specification of the training objective. Our models are more robust against the cross-entropy based attack (CE-PGD) than the pairwise distribution loss based attack (PA-PGD); thus, we report PA-PGD in all other plots.

Note that the phenomenon of vanishing gradients can occur in our models (as it can also occur in any cross-entropy based model). Thus, we decided to consider all attempted attacks, where the gradient is zero or zero for some of the pixels in the input image, as successful attacks because the attack might be mitigated by a lack of gradients. In Fig. 6, we compare conventional robustness to robustness with the additional Gradient-Non-Zero (GNZ) requirement. We emphasize that *we use the GNZ requirement also in all other plots.*

For evaluating the robustness against random noise, we use input noise with $\sigma \in \{0.25, 0.5, 0.75\}$ and clamp the image to the original pixel value range.

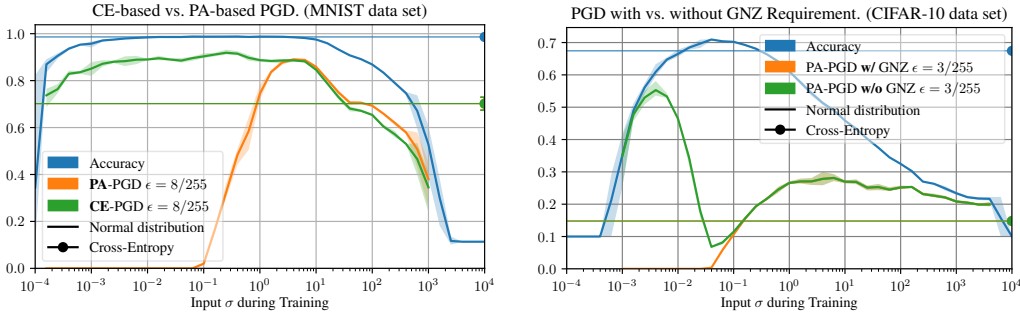

Figure 6: Ablations: *Left:* Softmax Cross-Entropy (CE) vs. pairwise distribution loss (PA) based PGD attack. Averaged over 3 runs. *Right:* Effect of the gradient-non-zero (GNZ) requirement. Averaged over 3 runs.

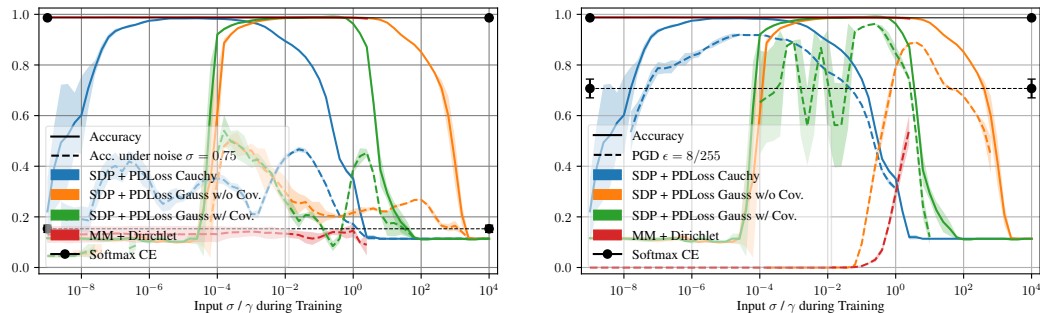

Figure 7: Robustness of CNNs on MNIST against Gaussian (left) and PGD adversarial noise (right). While Fig. 5 presents $\sigma = 0.5$ and $\epsilon = 16/255$, this figure presents $\sigma = 0.75$ and $\epsilon = 8/255$. Averaged over 3 runs.

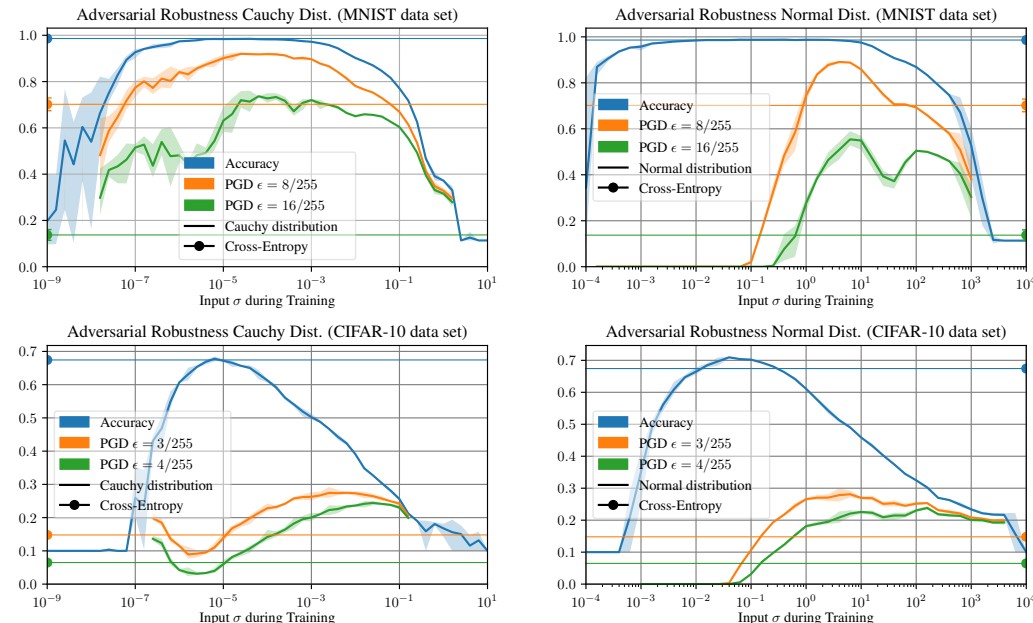

Figure 8: Accuracy and adversarial robustness under PGD attack. Left: Cauchy distribution. Right: Normal distribution *without* propagating covariances. Top: MNIST. Bottom: CIFAR-10. Averaged over 3 runs.

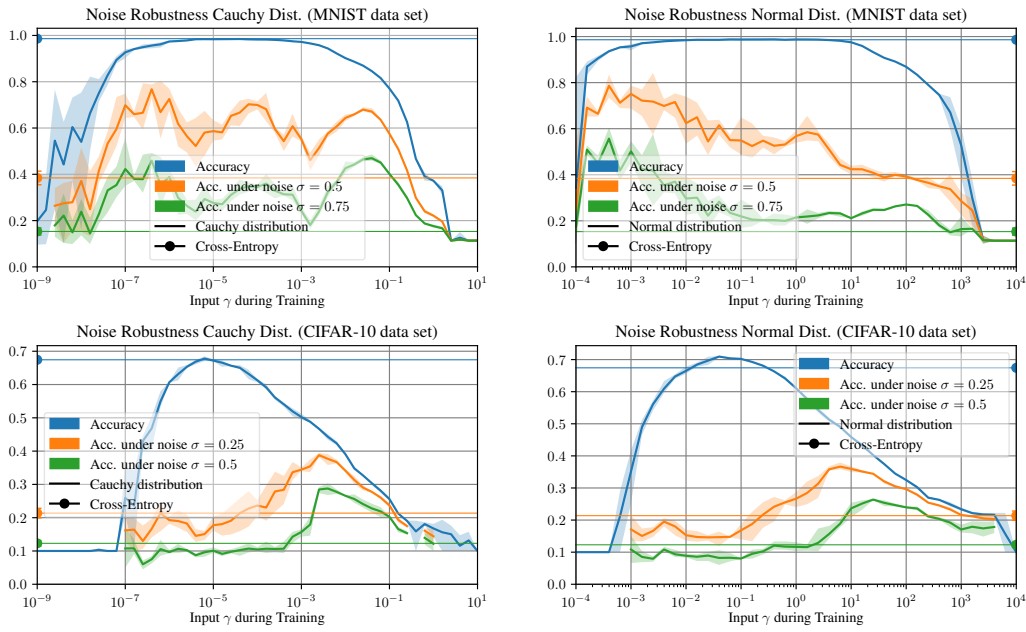

Figure 9: Accuracy and robustness against random normal noise. Left: Cauchy distribution. Right: Normal distribution *without* propagating covariances. Top: MNIST. Bottom: CIFAR-10. Averaged over 3 runs.

### B.5   MARGINAL PROPAGATION VISUALIZATION

We analyze the accuracy of a fully connected neural network (FCNN) where a Gaussian distribution is introduced in the $k$th out of 5 layers. Here, we utilize marginal SDP, i.e., propagation without covariances. The result is displayed in Figure 10, where we can observe that there is an optimal scale such that the accuracy is as large as possible. Further, we can see the behavior if the distribution is not introduced in the first layer, but in a later layer instead. That is, the first layers are conventional layers and the distribution is introduced after a certain number of layers. Here, we can see that the more layers propagate distributions rather than single values, the larger the optimal standard deviation is. The reason for this is that the scale of the distribution decays with each layer such that for deep models a larger input standard deviation is necessary to get an adequate output standard deviation for training the model. This is because we modeled the distributions without covariances, which demonstrates that propagating without covariances underestimates the variances.

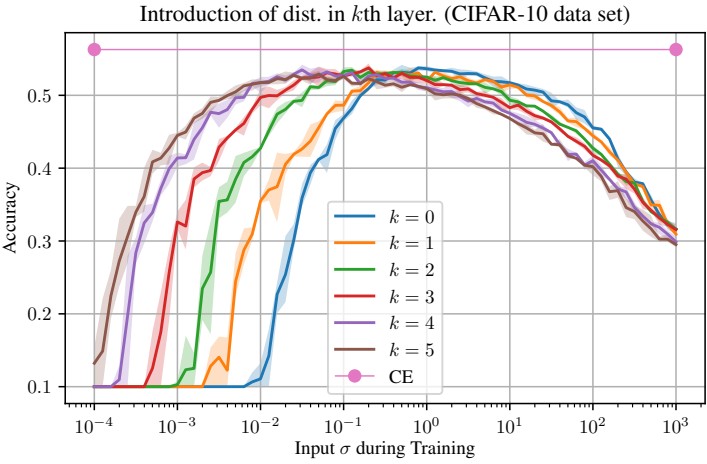

Figure 10: The normal distribution is introduced in the $k$th layer of a 5 layer fully connected network ($k \in \{0, 1, 2, 3, 4, 5\}$). Averaged over 3 runs.

## C   IMPLEMENTATION DETAILS

### C.1   SIMULATION OF PROPAGATING NORMAL DISTRIBUTIONS

We trained a fully-connected neural network with 1, 2, or 4 hidden layers and ReLU / Leaky-ReLU activations for 5000 epochs on the Iris data set [43] via the softmax cross-entropy loss. Here, each hidden layer has 100 hidden neurons. Results are averaged over 10 runs, and 10 inputs are propagated for each of these 10 runs. Across all methods of propagating distributions, each Monte Carlo baseline as well as model weights are shared.

### C.2   UCI UNCERTAINTY QUANTIFICATION

For the UCI uncertainty quantification experiment, we use the same hyperparameter and settings as [9]. That is, we use a network with 1 ReLU activated hidden layer, with 64 hidden neurons and train it for 5000 epochs. We perform this for 20 seeds and for a learning rate $\eta \in \{10^{-2}, 10^{-3}, 10^{-4}\}$ and weight decay $\in \{0, 10^{-3}, 10^{-2}, 10^{-1}, 1\}$. For the input standard deviation, we made a single initial run with input variance $\sigma^2 \in \{10^{-8}, 10^{-7}, 10^{-6}, 10^{-5}, 10^{-4}, 10^{-3}, 10^{-2}, 10^{-1}, 10^{0}\}$ and then (for each data set) used 11 variances at a resolution of $10^{0.1}$ around the best initial variance.

### C.3   APPLYING SDP TO PNNS

Here, we formalize how to apply Gaussian SDP to Gaussian PNNs. Let the PNN be described by $f_\mu(x)$ and $f_\Sigma(x)$. Conventionally, the PNN accordingly predicts an output distribution of $\mathcal{N}(f_\mu(x), f_\Sigma(x))$.

Using SDP, we extend the output distribution to

$$f_\mu(x) + \mathcal{N}(0, f_\Sigma(x)) + \mathcal{N}(0, \boldsymbol{J}(f_\mu)\boldsymbol{\Sigma}_x\boldsymbol{J}(f_\mu)^\top) = \mathcal{N}(f_\mu(x), f_\Sigma(x) + \boldsymbol{J}(f_\mu)\boldsymbol{\Sigma}_x\boldsymbol{J}(f_\mu)^\top) \quad (16)$$

where $\boldsymbol{\Sigma}_x$ is the input covariance matrix for data point $x$.

## C.4 SELECTIVE PREDICTION ON OUT-OF-DISTRIBUTION DATA

Here we use the same network architecture as for the robustness experiments in the next subsection (A CNN composed of two ReLU-activated convolutional layers with a kernel size of 3, a stride of 2, and hidden sizes of `64` and `128`; followed by a convolutional layer mapping it to an output size of 10) and also train it for 100 epochs, however, at a learning rate of $10^{-3}$. For training with our uncertainty propagation, we use an input standard deviation of $\sigma = 0.1$. For training with uncertainty propagation via moment matching and the Dirichlet loss as in [20], we use an input standard deviation of $\sigma = 0.01$, which is what they use in their experiments and also performs best in our experiment.

## C.5 GAUSSIAN AND ADVERSARIAL NOISE ROBUSTNESS

Here, our CNN is composed of two ReLU-activated convolutional layers with a kernel size of 3, a stride of 2, and hidden sizes of `64` and `128`. After flattening, this is followed by a convolutional layer mapping it to an output size of 10. We have trained all models for 100 epochs with an Adam optimizer, a learning rate of $10^{-4}$, and a batch size of 128.

## C.6 PROPAGATING WITHOUT COVARIANCES

The architecture of the 5-layer ReLU-activated FCNN is **784−256−256−256−256−10**.

## C.7 EVALUATION METRICS

### C.7.1 COMPUTATION OF TOTAL VARIATION IN THE SIMULATION EXPERIMENTS

We compute the total variation using the following procedure: We sample $10^6$ samples from the continuous propagated distribution. We bin the distributions into $10 \times 10 \times 10$ bins (the output is 3-dimensional) and compute the total variation for the bins. Here, we have an average of 1000 samples per bin. For each setting, we compute this for 10 input points and on 10 models (trained with different seeds) each, so the results are averaged over 100 data point / model configurations. Between the different methods, we use the same seeds and oracles. To validate that this is reliable, we also tested propagating $10^7$ samples, which yielded very similar results, but just took longer to compute the oracle. We also validated the evaluation methods by using a higher bin resolution, which also yielded very similar results.

### C.7.2 COMPUTATION OF MPIW

Given a predicted mean and standard deviation, the Prediction Interval Width (for $95\%$) can be computed directly. In a normal distribution, $68.27\%$ of samples lie within 1 standard deviation of the mean. For covering $95\%$ of a Gaussian distribution, we need to cover $\pm 1.96$ standard deviations. Therefore, the PIW is 3.92 standard deviations. We computed the reported MPIW on the test data set. This setup follows the setup by [9].

## C.8 MAXPOOL

For MaxPool, we apply Equation 1 such that $(\mu, \sigma) \mapsto (\max(\mu), \arg\max(\mu) \cdot \sigma)$ where $\arg\max$ yields a one-hot vector. The equation maxpools all inputs to one output; applying it to subsets of neurons correspondingly maxpools the respective subsets.

# D IMPLEMENTATIONS OF SDP: PSEUDO-CODE AND PYTORCH

In the following, we display a minimal Python-style pseudo code implementation of SDP.

```python
# Python style pseudo-code
model = ...        # neural network
data = ...         # inputs, e.g., images

y = model(data)
jac = jacobian(y, data)

output_mean = y
output_cov = jac @ jac.T * std**2
```

In the following, we provide an optimized PyTorch implementation of SDP in the form of a wrapper.

```python
# PyTorch SDP wrapper implementation.  (tested with PyTorch version 1.13.1)

class SDP(torch.nn.Module):
    def __init__(self, net, std=1., num_outputs=10, create_graph=True, vectorize=False):
        super(SDP, self).__init__()
        self.net = net
        self.std = std
        self.num_outputs = num_outputs
        self.create_graph = create_graph
        self.vectorize = vectorize
        # vectorize=True is only faster for large numbers of classes / small models

    def forward(self, x):
        assert len(x.shape) >= 2, x.shape  # First dimension is a batch dimension.

        # Two separate implementations:
        # the top one is for larger models and the bottom for smaller models.
        if not self.vectorize:
            x.requires_grad_()
            y = self.net(x)
            jacs = []
            for i in range(self.num_outputs):
                jacs.append(torch.autograd.grad(
                    y[:, i].sum(0), x,
                    create_graph=self.create_graph, retain_graph=True
                )[0])
            jac = torch.stack(jacs, dim=1).reshape(
                x.shape[0], self.num_outputs, np.prod(x.shape[1:])
            )

        else:
            e = torch.zeros(x.shape[0] * self.num_outputs, self.num_outputs).to(x.device)
            for out_dim_idx in range(self.num_outputs):
                e[x.shape[0] * out_dim_idx:x.shape[0] * (out_dim_idx+1), out_dim_idx] = 1.
            vjps = torch.autograd.functional.vjp(
                self.net, x.repeat(self.num_outputs, *[1]*(len(x.shape)-1)), v=e,
                create_graph=self.create_graph, strict=True
            )
            y = vjps[0][:x.shape[0]]
            jac = vjps[1].view(
                self.num_outputs, x.shape[0], np.prod(x.shape[1:])).transpose(0, 1)

        cov = torch.bmm(jac, jac.transpose(-2, -1)) * self.std ** 2

        return y, cov

# Usage example
model = ...        # neural network
data = ...         # inputs, e.g., images

sdp_model = SDP(model, std=0.1, num_outputs=10)
output_mean, output_cov = sdp_model(data)
```

