# OpenReview forum: "Uncertainty Quantification via Stable Distribution Propagation"
_ICLR.cc/2024/Conference — ICLR 2024 poster_

### Official Review · Reviewer_WExx · 2023-10-30

**Soundness:** 3 good
**Presentation:** 3 good
**Contribution:** 3 good
**Rating:** 8
**Confidence:** 4

**Summary:**

The authors propose an approach to propagate input uncertainty, i.e., noise in the input covariates
through neural network layers. The method is compared against moment-matching approaches in a wide series
of experiments.

**Strengths:**

- The approach can be applied to pre-trained networks with various activation functions and scales reasonably,
at least compared to common baselines.
- It can propagate both Gaussian, as well as, Cauchy distributions (and other $\alpha$-stable distributions)
- It can also estimate full covariance matrices instead of having to rely on marginal independence approximations
- The paper is well-written and explores a large range of experiments.

**Weaknesses:**

- The experiments are diverse but still rely on rather shallow network architectures. A somewhat deeper model,
ResNet-18 is only used for a runtime comparison.


### Minor
- The theoretical part of the paper only discusses uncertainty in terms of input/output/predictive uncertainties,
while in Section 4 suddenly the terms epistemic and aleatoric uncertainty appear without any prior introduction to them.
- Section 1, first paragraph. The abbreviation PNN has not been introduced at that stage of the paper
- UCI and Iris are never cited

**Questions:**

- How are the Table 1 and Table 3 baselines tables to be read if they refer to multiple baselines?
- How does the method perform with deeper networks? E.g., the ResNet-18 used in Sec4.3? It would be interesting to see
(i) a comparison of this depth where SDP is still viable, and (ii) a deeper setup to which only marginal MM and marginal SDP.

---

> ### Author Response · Authors · 2023-11-20
> **Response to Reviewer WExx**
>
> We thank you for the detailed review of our work and the extensive and helpful feedback.
> In the following, we address your comments and also uploaded a revision of the paper PDF.
> Please let us know if you have any remaining questions, concerns, and/or feedback.
>
>
>
> > "The experiments are diverse but still rely on rather shallow network architectures. A somewhat deeper model, ResNet-18 is only used for a runtime comparison." / "How does the method perform with deeper networks? E.g., the ResNet-18 used in Sec4.3? It would be interesting to see (i) a comparison of this depth where SDP is still viable, and (ii) a deeper setup to which only marginal MM and marginal SDP."
>
> We agree with this concern, and have therefore extended our evaluations to ResNet-18 (Table 33) as well as other deeper networks (10, 15, 20, 25 layers; Tables 36-39).
> We included the results in Supplementary Material D of the revision.
> We find that SDP outperforms moment matching in each case by a substantial margin, and even marginal SDP outperforms moment matching in most cases.
> Further, SDP also consistently outperformed DVIA for 10, 15, 20, and 25 layers.
>
> > "How are the Table 1 and Table 3 baselines tables to be read if they refer to multiple baselines?"
>
> In Table 1, we use marginal moment matching (aka. moment matching / assumed density filtering), which relies on two equations, which are shared among all cited references, i.e., each approach collapses to the same equation / implementation.
> We included multiple references to illustrate that the method is still commonly used in many forms and applications today, while also providing attribution to its original use by Frey & Hinton in the 90s.
>
> In Table 3, we followed "Tagasovska et al. [9]" in applying Deep Ensembles (Lakshminarayanan et al. [21]) to the UCI data uncertainty experiment.
> However, as the best choice of Deep Ensembles was indeed using only 1 PNN (which makes sense because it is a data uncertainty experiment), we refer to it as PNN, for which the original reference is [34], and the results are indeed the same for each method.
>
> > "The theoretical part of the paper only discusses uncertainty in terms of input/output/predictive uncertainties, while in Section 4 suddenly the terms epistemic and aleatoric uncertainty appear without any prior introduction to them."
>
> Thank you for pointing this out. We have replaced the terms aleatoric and epistemic uncertainties with data and model uncertainties, respectively, to be in harmony with the prior manuscript sections.
>
> > "Section 1, first paragraph. The abbreviation PNN has not been introduced at that stage of the paper."
>
> We have added the full form as Probabilistic Neural Network (PNN) on its first use.
>
> > "UCI and Iris are never cited."
>
> We have duly added the citations suggested by the referee. Additionally, we have also added citations for the MNIST and the EMNIST datasets used in the manuscript.

---

> > ### Comment · Reviewer_WExx · 2023-11-21
> >
> > Thank you for your answer. I keep my score of recommending acceptance.

---

### Official Review · Reviewer_tarK · 2023-10-31

**Soundness:** 3 good
**Presentation:** 2 fair
**Contribution:** 2 fair
**Rating:** 6
**Confidence:** 4

**Summary:**

In this paper, the authors introduce a method for stable probability distribution propagation through neural networks. They specifically consider the case of ReLU neural networks and demonstrate that their suggested local linearization is the optimal approximation based on the total variation distance.

The authors focus on two specific examples: propagating Gaussian and Cauchy distributions. Additionally, they present experiments that show the benefits of their proposed method.

**Strengths:**

1. The paper is well-structured and easy to follow.
2. The set of experiments is quite diverse (however see Weaknesses section)
3. The authors' focus on specific examples, like the propagation of Gaussian and Cauchy distributions, adds clarity to their explanations.
4. The introduction of local linearization as an optimal approximation based on the total variation distance is a novel approach.
5. The paper has a good balance between theoretical and practical sides, making it useful for both researchers and practitioners.

**Weaknesses:**

My main concern is the small number of layers used in the tests. In practice, most times, we use networks with up to hundreds of layers, but this paper only used up to 6 layers. So, it's hard to know how well the method will work for bigger networks. Especially not clear, how accurate and reliable will be the propagation of covariance in the case of Gaussian.

---

There are other concerns:


- There are typos:
1. The term "PNN" is used without explaining it first in the Introduction.
2. On Page 5, the phrase "the the" is repeated by mistake.

- Experiments:
1. Table 2 used very small networks with only 4 layers. The biggest networks in the extra section have only 7 layers. This doesn't help us understand how the method works on big (practical) networks.
2. There's no code provided.

- References:
[16] - No authors were mentioned in the paper.

**Questions:**

1. How to choose the distribution which to propagate?
2. Figure 3 -- which measure of uncertainty was used?

---

> ### Author Response · Authors · 2023-11-20
> **Response to Reviewer tarK**
>
> We thank you for the detailed review of our work and the extensive and helpful feedback.
> In the following, we address your comments and also uploaded a revision of the paper PDF.
> Please let us know if you have any remaining questions, concerns, and/or feedback.
>
> > "My main concern is the small number of layers used in the tests. [...]", "Table 2 used very small networks with only 4 layers. The biggest networks in the extra section have only 7 layers. This doesn't help us understand how the method works on big (practical) networks."
>
> To address this concern, we have run additional experiments. In particular, we have considered 10, 15, 20, and 25 layers in Tables 36-39.
> Moreover, we have run a similar experiment for ResNet-18 and report the results in Table 33.
> We find that SDP (i.e., with covariance matrix) outperforms moment matching in each case by a substantial margin, and even marginal SDP outperforms moment matching in most cases.
>
> > "There's no code provided."
>
> We have included a PyTorch implementation of SDP in Supplementary Material E, and we will be publishing the code upon publication of the paper, also including marginal SDP and experiments.
>
> > "How to choose the distribution which to propagate?"
>
> The knowledge of the input uncertainty distribution would be based on the user's prior knowledge of the problem. In a case where such domain knowledge is absent, the user can choose the maximum entropy prior distribution (e.g., Gaussian) and estimate its parameters via cross validation.
>
> > "Figure 3 -- which measure of uncertainty was used?"
>
> The plot displays the means and standard deviations of the risk-coverage between all 10 seeds. Please elaborate and let us know in case this does not answer your question.
>
> > PNN, the the, ref 16
>
> Thank you for pointing out the typos, we have fixed each of them.

---

> > ### Comment · Reviewer_tarK · 2023-11-22
> >
> > I would like to thank authors for their answers.
> >
> > Given the new experiments for deeper networks and the well-addressed concerns, I am willing to increase the score.

---

### Official Review · Reviewer_iv48 · 2023-11-01

**Soundness:** 3 good
**Presentation:** 2 fair
**Contribution:** 2 fair
**Rating:** 6
**Confidence:** 3

**Summary:**

This paper proposes an algorithm approximating the output distribution of a neural network under input uncertainty. Specifically, the problem is approximating the distribution $f(x + \epsilon)$ where $\epsilon$ is a known distribution. For distributions with independent covariances, they propose transforming the input covariance layer-by-layer by multiplying with the derivative of the activation functions. For the computationally more challenging case of propagating the full covariance, they propose linearizing the entire network and transforming the input covariance by multiplying with the Jacobian of the network. Empirically, they demonstrate the methods achieve competitive performance on uncertainty quantification and out-of-distribution selective prediction.

**Strengths:**

- This a simple method that leads to improved performance in some settings.
- Theorem 1 guarantees the simple formula propagating distributions is optimal in the total variation distance.

**Weaknesses:**

- Theorem 1 has limited applicability. It may not even be applicable to some of the authors' own methods.
    - As far as I can tell, Theorem 1 only works for univariate Gaussian distributions. Thus, this optimal approximation result does not apply to propagating the full covariance (including the SDP proposed in this paper).
    - As a compromise for computation efficiency, Section 3.3 proposes linearizing the entire network, including the ReLU activation. The covariance of the output distribution is computed by simply multiplying the Jacobian on the input covariance. Thus, the optimal approximation result in Theorem 1 does not apply either.
    - If the above are true, it is better to be upfront about it in the main text and state explicitly which method does Theorem 1 apply.
- Some evaluation metrics are not easy to compare. For example, Table 3 reports the prediction interval coverage probability (PICP, the higher the better) and the mean prediction interval width (MPIW, the lower the better). Each method has different PICP and MPIW which makes it hard to distinguish which one is the best. Since the network outputs a probability distribution, it might be better to report the test log likelihood, which captures the prediction accuracy and the length of the prediction interval simultaneously in a single metric.

**Questions:**

- Can this method be applied in randomized smoothing (e.g., Cohen et al. 2019) in adversarial robustness? Computing the output distribution in this paper is very similar to smoothing the input with random noise. I wonder if the technique can be used to speed up randomized smoothing, which currently uses Monte Carlo estimates.

---

> ### Author Response · Authors · 2023-11-20
> **Response to Reviewer iv48**
>
> We thank you for the detailed review of our work and the extensive and helpful feedback.
> In the following, we address your comments and also uploaded a revision of the paper PDF.
> Please let us know if you have any remaining questions, concerns, and/or feedback.
>
> > Thorem 1
>
> First, we would like to remark that Theorem 1 (+Corollary 2 and Corollary 3) apply generally to $\alpha$-stable distributions.
> You are indeed correct that the Theorem 1 applies to univariate distributions. This is similar to how a result of "moment matching is an optimal estimator of the moments" is also only applicable in the univariate case. For this reason, we included extensive numerical studies to evaluate the behavior empirically (Tables 1-2, 6-31, 33, 36-39).
>
> > "As a compromise for computation efficiency, Section 3.3 proposes linearizing the entire network, including the ReLU activation. [...]"
>
> We would like to clarify this confusion. Section 3.3 is not a compromise for computation efficiency. Instead, it shows how to efficiently compute SDP exactly and without introducing an approximation here. We added a clarification to the paper.
>
> > "Some evaluation metrics are not easy to compare. For example, Table 3 reports the prediction interval coverage probability (PICP, the higher the better) and the mean prediction interval width (MPIW, the lower the better). Each method has different PICP and MPIW which makes it hard to distinguish which one is the best. Since the network outputs a probability distribution, it might be better to report the test log likelihood, which captures the prediction accuracy and the length of the prediction interval simultaneously in a single metric."
>
> The objective of the experiment in the table reported was to evaluate the sharpness of best calibrated prediction intervals from different UQ methods. This test has been used by prior authors. The negative log likelihood would not inform about the calibration of the prediction interval for this comparison.
>
> That said, we completely agree with the reviewer that the test negative log likelihood is the metric that captures both accuracy and width of prediction intervals simultaneously. As instructed by the reviewer, we have added test negative log likelihood comparisons for the experiment in Table 3 in the Supplementary Material (Table 34).
>
> > "Can this method be applied in randomized smoothing (e.g., Cohen et al. 2019) in adversarial robustness? Computing the output distribution in this paper is very similar to smoothing the input with random noise. I wonder if the technique can be used to speed up randomized smoothing, which currently uses Monte Carlo estimates."
>
> Yes, SDP can indeed be applied in adversarial robustness, in particular, our Supplementary Material B.3 contains a range of evaluations in this regard.

---

> > ### Comment · Reviewer_iv48 · 2023-11-23
> >
> > I thank the authors' clarification on the content in section 3. I remain positive about the paper but I will withstand my current score.

---

### Official Review · Reviewer_ojgP · 2023-11-02

**Soundness:** 3 good
**Presentation:** 3 good
**Contribution:** 3 good
**Rating:** 8
**Confidence:** 3

**Summary:**

The paper proposes a novel method for propagating distributions through layers of neural networks to estimate uncertainty of predictions arising through uncertainty in data. The paper proposes to use local linearisation that is proved in the paper to provide the optimal approximation in terms of total variance for RELU non-linearity. Moreover, empirically the paper also shows that the method works for other types of non-linearities.

**Strengths:**

* A novel method for an important problem of uncertainty quantification, and namely, via distribution propagation
* The idea of local linearisation elegantly works for ReLU as a simple transform of mean and variance
* Theoretical evaluation of the proposed method
* Extensive evaluation from different angles including robustness to noise and adversarial attacks

**Weaknesses:**

* Comparison to other models seems a bit limited. DVIA is only used in a toy example due its prohibited cost, Section 4.1 only has 2 baselines, none of which include, for example, Bayesian neural networks.
* Presentation sometimes left me unclear what was intended to be said. For example, I didn't understand the setting of pairwise probability estimation for classification in Section 3.5

Specific comments:
1.	End of the first paragraph. PNN is not defined.
2.	Section 1. What is Y?
3.	Ref 16 does not have authors
4.	I believe Section 2 is missing an important, one of the first works in the area. Hernández-Lobato, J.M. and Adams, R., 2015. Probabilistic backpropagation for scalable learning of Bayesian neural networks. In International conference on machine learning (pp. 1861-1869).
5.	Table 1. It is not very clear what marginal moment matching with several references means. Did the authors try all these references and report the best results among them?
6.	Elaboration of the idea of pairwise probability of correct classification is required in Section 3.5
7.	Section 4.1. It would be good to have description of MPIW similar to PICP


Minor:
1.	Section 3.4. End of the first paragraph. “wrt. the the” -> “wrt. the”

**Questions:**

Could you please elaborate on learning with SDP for the classification case? I.e. on "we propose computing the
pairwise probability of correct classification among all possible pairs in a one-vs-one setting". What is the "pairwaise probability of correct classification"? How does one-vs-one setting works for multiclass classification problems?

---

> ### Author Response · Authors · 2023-11-20
> **Response to Reviewer ojgP**
>
> We thank you for the detailed review of our work and the extensive and helpful feedback.
> In the following, we address your comments and also uploaded a revision of the paper PDF.
> Please let us know if you have any remaining questions, concerns, and/or feedback.
>
> > "Comparison to other models seems a bit limited. DVIA is only used in a toy example due its prohibited cost, Section 4.1 only has 2 baselines, none of which include, for example, Bayesian neural networks."
>
> Regarding uncertainty propagation experiments, DVIA and marginal moment matching are to our knowledge the only existing methods for propagating input uncertainties through neural networks. If applicable, please point us to a respective reference for any other propagation methods.
>
> Regarding Section 4.1, we remark that Bayesian neural networks are for estimating model uncertainty and not for estimating data uncertainty. Accordingly, their applicability in this experiment is limited. Further, MC Estimate is a quite strong baseline, as it is an unbiased estimator and we remark that the column PNN includes deep ensembles, which only collapse to PNN because deep ensembles performs best using only one model (as it is a data uncertainty experiment).
>
> > Pairwise probability estimation for classification in Section 3.5 / "What is the "pairwaise probability of correct classification"? How does one-vs-one setting works for multiclass classification problems?" / "6. [...]"
>
> Historically, one-vs-one settings have had a long tradition of being applied to multi-class classification problems.
> In particular, the ground truth class score is compared to each of the false class scores, a loss is computed for each of these comparisons, and then these losses are averaged.
> Historically, this gave rise to the multivariate logistic losses, which later was improved to softmax-based losses.
> The "pairwaise probability of correct classification" is defined in Equations 7-9, i.e., it is the probability that the score for the true class is greater than the score for a respective false class. We clarified the description in the paper.
>
> > "1. End of the first paragraph. PNN is not defined"
>
> Thank you, we have added the full form as Probabilistic Neural Network (PNN) on its first use.
>
> > "2. Section 1. What is Y?"
>
> Y is the true output distribution, which is defined via $f(x+\epsilon)\sim Y$. We have clarified this in the revision.
>
> > "3. Ref 16 does not have authors"
>
> Thank you, we fixed reference 16.
>
> > "4. I believe Section 2 is missing an important, one of the first works in the area. Hernández-Lobato, J.M. and Adams, R., 2015. Probabilistic backpropagation for scalable learning of Bayesian neural networks. In International conference on machine learning (pp. 1861-1869)."
>
> Thank you, we included it in the revised manuscript.
>
> > "5. Table 1. It is not very clear what marginal moment matching with several references means. Did the authors try all these references and report the best results among them?"
>
> Moment matching, assumend density filtering, etc., all use the same equation for propagating moments, and therefore all references collapse to the same equation / method.
> We refer to it as marginal moment matching because we also explicitly discuss full moment matching (which is computationally intractible, but DVIA provides an approximation to full moment matching).
>
> > "7. Section 4.1. It would be good to have description of MPIW similar to PICP"
>
> We agree and have added a description of the MPIW, in the same location as the PICP's.
>
> > "Minor: 1. Section 3.4. End of the first paragraph. “wrt. the the” -> “wrt. the"
>
> Thanks, we have corrected the typo.

---

> > ### Comment · Reviewer_ojgP · 2023-11-21
> >
> > Thank you very much for your detailed answer. You have clarified most of my concerns. I would suggest adding clarification about baselines (the lack of them, deep ensembles collapsing to PNN, MC Estimate being a strong baseline, more emphasis that BNN are not applicable for data uncertainty) into the manuscript as it was confusing for other reviewers as well, so there is a high probability it would be confusing for some of the readers.
> >
> > I am increasing my score as the response has cleared my initial concerns.

---

### Official Review · Reviewer_e2L6 · 2023-11-04

**Soundness:** 3 good
**Presentation:** 2 fair
**Contribution:** 2 fair
**Rating:** 6
**Confidence:** 4

**Summary:**

The authors propose an uncertainty quantification method that analytically propagates uncertainty around the data distribution throughout the network to produce better uncertainty estimates for the output. The key technique used by the authors involves local linearization.

**Strengths:**

- The method proposed by the authors is simple and easy-to-use
- Some elements of the presentation are polished, including the figures and the motivation
- The method can be combined with other UQ approaches such as PNNs and thus improve the performance of a wide range of methods
- The experimental results suggest that the method has the potential to improve the performance of existing approaches

**Weaknesses:**

- I think some parts of the presentation can be further improved. In particular, while the methods section explains how to propagate uncertainties between layers, it would help to explain how the method is applied to an entire neural net, i.e., what is the recipe needed to go from the formulas in the methods section to getting uncertainties from a general neural network. Perhaps an algorithm float could be helpful for that. Perhaps pseudocode in the appendix would help. Right now, it requires the reader to think for themselves.
- The theory only works for certain types of activations and distributions. This should be acknowledged and discussed a more clearly in the paper.
- The computational considerations remain somewhat unclear to me. In the experiments section, training a resnet appears to take 30x longer. There a simpler mar-SDP method that is faster, but it is unclear to me where it is evaluated and what are the pros/cons of going between SDP and mar-SDP. Also, the methods section seems to suggest that computational cost is not a big concern (saying things like "the Jacobian easy to compute"), but it's not clear to me why that is the case, especially given the empirical results.
- The experiments are on very simple datasets (UCI, MNIST). I wonder if the computational cost is something that makes it hard to test on bigger datasets.
- The experimental results are not entirely convincing. On the UCI table, while directionally things look right, the numbers are often within error bars. The baselines are also not very convincing: in the UCI table, the MC method is known to not be very good in practice. There are probably stronger baselines such as deep ensembles (Lakshminarayanan et al., NeurIPS 2017), or post-hoc recalibration (Kuleshov et al., ICML 2018), or more recent methods for quantile estimation (Si et al, ICLR 2022). (Perhaps SDP can be combined with these methods)? The second experiment has even fewer baselines, and the table is missing error bars (although they seem to be in the figure).

**Questions:**

- How is the method applied end-to-end to a neural net?
- Can the authors provide a more detailed discussion of computational considerations?
- What are the effects of assumptions on the distribution and the activation type?
- How does the method compare to stronger baselines?
- Can the method be combined with other UQ methods?

---

> ### Author Response · Authors · 2023-11-20
> **Response to Reviewer e2L6 [1/2]**
>
> We thank you for the detailed review of our work and the extensive and helpful feedback.
> In the following, we address your comments and also uploaded a revision of the paper PDF.
> Please let us know if you have any remaining questions, concerns, and/or feedback.
>
> > "while the methods section explains how to propagate uncertainties between layers, it would help to explain how the method is applied to an entire neural net, i.e., what is the recipe needed to go from the formulas in the methods section to getting uncertainties from a general neural network. Perhaps an algorithm float could be helpful for that. Perhaps pseudocode in the appendix would help. Right now, it requires the reader to think for themselves." / "How is the method applied end-to-end to a neural net?".
>
> Thank you for the suggestion. In our revision, we have now included a pseudo-code implementation as well as an optimized PyTorch implementation of SDP in Appendix E to demonstrate how the method is applied end-to-end to a neural net.
> We remark that for marginal SDP (or marginal moment matching) each layer of the neural network requires separate implementations as these do not allow for end-to-end implementations, which we will provide upon publication.
>
> > Theory / "What are the effects of assumptions on the distribution and the activation type?"
>
> Regarding the theoretical result, yes, it only applies to ReLU non-linearities for stable distributions, but we remark that the approach performs well on various other non-linear functions (Leaky-ReLU, sigmoid, SiLU, GeLU, MaxPool) empirically as shown, e.g., in Tables 18-31 and Table 33.
> We have included an additional statement in the introduction regarding the restriction on input distributions.
>
> > "The computational considerations remain somewhat unclear to me. In the experiments section, training a resnet appears to take 30x longer. There a simpler mar-SDP method that is faster, but it is unclear to me where it is evaluated and what are the pros/cons of going between SDP and mar-SDP. Also, the methods section seems to suggest that computational cost is not a big concern (saying things like "the Jacobian easy to compute"), but it's not clear to me why that is the case, especially given the empirical results." / "Can the authors provide a more detailed discussion of computational considerations?"
>
> Regarding the computational considerations, the Jacobian is relatively easy to compute in comparison to other approaches that either require exponential complexity (computing it exactly) or cubed complexity in the number of activations in the neural network (DVIA).
> The computational cost of computing the Jacobian is typically proportional to m propagations where m is the number of classes / output dimension.
> The reason for why it was 25 times slower than regular training was only an implementation reason.
> With our current implementation and a more recent PyTorch version than originally, SDP takes only 4.3 times longer compared to the default on ResNet-18 with CIFAR-10.
>
> > "The experiments are on very simple datasets (UCI, MNIST). I wonder if the computational cost is something that makes it hard to test on bigger datasets."
>
> The reason for us to limit the uncertainty propagation experiments primarily to UCI is actually the large computational cost of obtaining the ground truth oracles (which we do by propagating large numbers of samples).
> We now have uncertainty propagation experiments for ResNet-18 on CIFAR-10 (Table 33), where computing the oracle took many hours compared to the distribution propagation methods, which took seconds.

---

> ### Author Response · Authors · 2023-11-20
> **Response to Reviewer e2L6 [2/2]**
>
> > Experimental Results / "How does the method compare to stronger baselines?"
>
> Regarding, "MC Estimate", we want to point out that this is not "MC Dropout", which may have caused confusion.
> With "MC Estimate", we refer to the procedure of sampling k inputs from the input distribution, propagating each sample through the network, and then computing mean and covariance matrix based on these samples. This is quite a strong method as it is an unbiased estimator.
> We have added a clarification in the paper for Table 3.
> Further, we would like to point out that, for data uncertainty, the best deep ensemble is an ensemble with only 1 model, which makes it exactly equivalent to a PNN (the ensembling helps with model uncertainty, but not for data uncertainty).
> Indeed, we ran deep ensembles (with an additional option of only 1 model), and found that 1 model performs best, which is why we labelled it PNN, but still cite the original PNN as well as (Lakshminarayanan et al., NeurIPS 2017) at this point in Table 3.
> This aligns with the results in Tagasovska et al. (2019), where we adopted the experiment from.
> We included references to the methods you mentioned.
> For post-hoc recalibration, we want to remark that recalibration is orthogonal to the choice of the "base UQ method", which means it could be applied on top of each of the methods in Table 3, including out methods.
>
> Regaring the experiments in Table 4, we have now included a Table with standard deviations (Table 35).
> The experiment has a number of baselines, in particular, softmax entropy, dirichlet outputs / distributions, moment matching, and orthonormal certificates, as well as combinations of these methods.
>
> > "Can the method be combined with other UQ methods?"
>
> Beyond the combination of SDP with PNNs, SDP could be combined also with other UQ methods that complement each other.
> For example, the hybrid SDP+PNN method with ensembling and adversarial training along the lines of Deep Ensembles to account for both data and model uncertainties.
> Further, post-hoc recalibration could also be combined with SDP.

---

### Author Response · Authors · 2023-11-20

Dear reviewers and AC,

We want to thank you very much for your time and for the reviews.
We provide a response to each of your concerns below.

Moreover, we have included a revision of the PDF with the following major additions to address the reviewers' concerns:

* Experiements with ResNet-18 (Table 33)
* Experiments with 10, 15, 20, 25 layered networks (Tables 36-39)
* Negative Log Likelihood metric corresponding to Table 3 (Table 34)
* Pseudo-code and PyTorch implementation of SDP (Appendix E)

For your convenience, we marked the changes throughout the paper in blue, and have added the new tables and code on the last 4 pages of the appendix.

Please let us know if you have any other questions or concerns regarding our paper or response.

Best regards,

The Authors

---

### Meta-Review · Area_Chair_StVv · 2023-12-07

**Metareview:**

The paper proposes a new method to propagate probability distributions (Gaussian, Cauchy, ...) through a neural network. The paper shows that the proposed method is optimal for each layer in the total-variation sense.  Most previous works rely on assumed density filtering (moment matching), but this paper uses a locally optimal total-variation matching which is faster and works for a broader class of distributions (e.g., Cauchy distributions which do not have a finite second moment).

The strength of the paper is the clarity of the writing, the simplicity of the proposed approach and the encouraging numerical experiments. The method outperforms moment-matching approaches, in particular, the one used in Deterministic Variational Inference. In some cases, the method is even more accurate than using 100 Monte-Carlo samples.

The main weakness is the small-scale experimental evaluation, which has been partially addressed in the rebuttal.

Overall, I recommend acceptance of the paper as a poster.  For the final version, I encourage the authors to take the reviewers' feedback into account.

**Justification For Why Not Higher Score:**

Larger-scale experiments and more downstream applications would be needed to understand in which situations the method works well an why.  Theoretical results only show local optimality for ReLU, but global optimality or error bounds are not discussed. Therefore, it remains unclear why or when the approach works well.  For an oral or spotlight recommendation, I would have liked to see some of the aspects addressed.

**Justification For Why Not Lower Score:**

The paper is well-written, easy to read and proposes a simple and useful method that works well on some small-scale experiments. I believe that the paper is useful to the community. Therefore, I recommend acceptance.

---

### Decision · Program_Chairs · 2024-01-16

Accept (poster)